# Epithelial cell competition is promoted by signaling from immune cells

Yilun Zhu [1,2], Zeba Wunderlich [3,4] & Arthur D. Lander [1,2,5,6] ✉

In epithelial tissues, juxtaposition of cells of different phenotypes can trigger cell competition, a process whereby one type of cell drives death and extrusion of another. During growth and homeostasis, cell competition is thought to serve a quality control function, eliminating cells that are "less fit". Tissues may also attack and eliminate newly arising tumor cells, exploiting mechanisms shared with other instances of cell competition, but that differ, reportedly, in the involvement of the immune system. Whereas immune cells have been shown to play a direct role in killing tumor cells, this has not been observed in other cases of cell competition, suggesting that tissues recognize and handle cancer cells differently. Here, we challenge this view, showing that, in the fruit fly *Drosophila*, innate immune cells play similar roles in cell killing during classical cell competition as in eliminating tumors. These findings suggest that immune suppression of cancer may exploit the same mechanisms as are involved in promoting phenotypic uniformity among epithelial cells.

The importance of immune cell function in the initiation and progression of cancer has attracted growing attention, fueled in part by the success of immune-based cancer therapies[1]. Although such therapies are primarily aimed at stimulating the adaptive immune system—T cells in particular[2]—there is longstanding evidence that innate immune cells—especially NK cells and cells of the myeloid/macrophage lineage—also matter greatly in cancer[3]. Innate immune cells have been implicated not only in responses to established tumors, but also in tumor surveillance—the identification and elimination of cancer cells when they first arise[4–13]. Given that innate immune cells participate in a broad range of events in tissue growth, morphogenesis, and differentiation[14,15], an obvious question is what relationship, if any, exists between the normal physiological functions of innate immune cells and their responses to cancer cells.

The fruit fly, *Drosophila melanogaster*, provides a useful platform for addressing this question because, like other insects, it lacks an adaptive immune system, relying exclusively on cells known as hemocytes that exhibit features of NK cells and macrophages[16–18]. *Drosophila* displays robust tumor suppression, which we define as recognition and elimination of tumor cells shortly after they first arise.

This process is illustrated by the fact that small mosaic clones that are mutant for genes such as *scribble*, *dlg1* and *l(2)gl*, when created in the imaginal discs of larvae, are efficiently killed and actively extruded, whereas large, contiguous groups of the same mutant cells grow uncontrollably[19–26]. This process depends both on the direct actions of adjacent wildtype epithelial cells, as well as the actions of hemocytes that get recruited to sites where mutant and wildtype cells are juxtaposed[25,27]. Hemocytes have been shown to play at least two roles in this context: they secrete the pro-apoptotic ligand Eiger, the sole tumor necrosis factor (TNF) ortholog in *Drosophila*, and they carry out phagocytosis of dead cells[25,28,29]. Eiger activates JNK signaling and triggers apoptosis in a wide variety of cells[30,31], and although hemocytes are not the only source of Eiger available to imaginal discs[24,32], experiments show that hemocyte-derived Eiger is required for tumor suppression.

Many aspects of the above response are shared with a phenomenon known as "cell competition", which occurs when cells with slightly differing growth phenotypes are juxtaposed, typically within an epithelium. Although cell competition was first described in *Drosophila* imaginal discs[33,34], it has been documented in multiple

[1]Department of Developmental and Cell Biology, University of California, Irvine, Irvine, CA 92697, USA. [2]Center for Complex Biological Systems, University of California, Irvine, Irvine, CA 92697, USA. [3]Department of Biology, Boston University, Boston, MA 02215, USA. [4]Biological Design Center, Boston University, Boston, MA 02215, USA. [5]Department of Biomedical Engineering, University of California, Irvine, Irvine, CA 92697, USA. [6]NSF-Simons Center for Multiscale Cell Fate Research, University of California, Irvine, Irvine, CA 92697, USA. ✉e-mail: adlander@uci.edu

vertebrate contexts[35–43]. The hallmark of cell competition is that one group of cells (so-called "losers") responds to being juxtaposed with another ("winners") by reducing its rate of proliferation or increasing its rate of cell death, allowing the winners to expand and take over previously occupied territory[33,34,44–53]. The phenomenon is often described as serving an epithelial quality control function, giving healthier cells the opportunity to eliminate less "fit" neighbors[54,55].

In *Drosophila*, similarities between tumor suppression and other examples of cell competition include a requirement for the juxtaposition of phenotypically different cells; involvement of Eiger, JNK signaling, and suppression of Hippo signaling; and a process of cell elimination that involves cell killing and active extrusion from the epithelium; these similarities have led many authors to refer to tumor suppression as a "form" of cell competition[27,35,56].

Yet an important difference between traditional cell competition and tumor suppression concerns the roles attributed to hemocytes. It has been reported that hemocytes do not accumulate at sites of cell competition[57], that JNK signaling is not strictly required for cell competition[58], and that the primary function of hemocytes in cell competition is to phagocytose the "corpses" of dead cells[28,29,59]. Although cell engulfment is an essential event—winner cells seem to need "space" to be cleared to enable proliferative expansion—the generally accepted view is that immune cell function in cell competition is largely secondary, and that the active process of cell recognition and elimination occurs primarily among competing epithelial cells themselves[59–63].

Here we revisit this conclusion, carrying out experiments aimed at visualizing and perturbing hemocytes during classical assays of cell competition in *Drosophila* imaginal discs, i.e., involving the generation of mosaic clones that over- or under express Myc relative to neighboring cells, as well as mosaic clones that are wildtype in a background heterozygous for a Minute mutation (a mutation in a ribosomal protein gene). We show that hemocytes are indeed recruited during these assays; cell death is not required for such recruitment; hemocytes do play a role in cell killing; and production of Eiger by hemocytes is required, at least in part, for competition.

These results suggest that the suppression and elimination of tumor cells in *Drosophila*—and potentially in other organisms as well—need not invoke any "cancer-specific" mechanisms other than those already in place to respond to phenotypic mismatches within epithelia. To this end, efforts to improve tumor surveillance in human populations might be well served by a deeper understanding of the more general process of cell competition—in particular, how winner and loser fates get assigned, and how innate immune cells are attracted to sites where cell phenotypes clash.

## Results

### Hemocytes are essential for cell competition

To generate cell competition in imaginal discs (Fig. 1), we initially created "twin-spot" clones differing either in dosage of *dMyc* or the presence of a *Minute* mutation (Fig. S1). In the first case, the heat-shock inducible flippase (hs-Flp)/flippase recognition target (FRT) system was used to generate pairs of clones containing either two or zero extra copies of *dMyc*, in a background of a single extra copy of *dMyc*[64]. Clones with 2x *dMyc* were marked by a lack of GFP, and those without extra *dMyc* by two copies of GFP (Figs. 1B and S1A). In the second case, the hs-Flp/FRT system was used to generate, in a background heterozygous for a *Minute* mutation, LacZ-expressing *Minute*-homozygous clones and LacZ-negative wildtype clones[65] (Figs. 1E and S1B). For comparison, wildtype control clones were generated in similar fashion using the same marker genes, but without differences in *dMyc* dosage (Fig. 1A) or *Minute* genotype (Fig. 1D). Imaginal discs were dissected at various times after clone induction (ACI) and examined for GFP or LacZ expression. The total area occupied by cells of each genotype was assessed, and the success of cell competition quantified, as described

in Methods, as a ratio in each disc of the areas occupied by cells of the no-GFP versus 2x-GFP genotypes (Fig. S2A, B) or the areas occupied by cells of the wildtype (LacZ-) versus *Minute*-containing (LacZ + ) genotypes (Fig. S2F).

With *dMyc* clones, the ratio between 2x *dMyc* and wildtype areas was about two, whereas in control experiments (in which clones differed only in GFP, and regardless of whether or not the background contained a single extra copy of *dMyc*) it was one, indicating that GFP itself confers no growth advantage, whereas extra Myc turns cells into "supercompetitors"[50,51].

Similarly, clones lacking the *Minute* mutation occupied about 4 times the area, on average, as *Minute* cell clones (Fig. 1H). In control experiments (clones differing only in LacZ expression), the ratio of LacZ+ to LacZ- areas was about 2, demonstrating that wildtype cells efficiently outcompete *Minute* cells (as homozygous Minute cells do not survive, the area ratio in these experiments compares wildtype cells (LacZ-) with the heterozygous background (LacZ + )).

To ablate hemocytes, we drove expression of the proapoptotic *UAS-hid* transgene under the control of the hemocyte-specific driver *hml-Gal4*[66]. Expressing *UAS-hid* efficiently eliminated detectable hemocytes from discs (Fig. S3) and did not result in any detectable developmental delay either in a wild-type background or in a background heterozygous for a *Minute* mutation (Fig. S4). We generated *dMyc* (Fig. 1C) and *Minute* (Fig. 1F) clones on this background and examined wing discs. In both cases, area ratios were markedly and statistically significantly restored toward control values. With *dMyc* clones there appeared to be a total blockade of cell competition (Fig. 1G). With *Minute* clones, cell competition was also markedly rescued, although not quite as completely as was the case with *dMyc* clones (Fig. 1H; Fig. S5). These data indicate that both *dMyc*-associated and *Minute*-associated cell competition require the presence of hemocytes.

### Hemocytes are recruited to sites of cell competition

As *hml-Gal4* expresses in hemocytes throughout the animal, a requirement for hemocytes in larval cell competition does not itself indicate where in the larva hemocytes act. As others have described[29,57,67,68], discs are not particularly rich in hemocytes. Indeed, in wild-type larvae in which *hml-Gal4* was used to drive *UAS-RFP*, we observed only a few labeled cells per disc (Fig. 2A, A' and D); similar results were obtained using a hemocyte-specific *srpHemo-3xmCherry* transgene[69] (Fig. S7).

In contrast, when *dMyc* twin-spot clones were generated in either of these backgrounds, a large increase in the numbers of hemocytes was observed, from as early as 16 h ACI up through 72 h (Figs. 2A–D and S6, S7). Hemocytes were detected on the external faces of discs, both columnar and peripodial, and only very rarely in the interior, luminal space (Fig. S8), consistent with idea that hemocytes deposit onto the disc from the surrounding hemolymph. We observed that accumulations of labeled cells were often associated with folds, perhaps because they served to protect attached cells from being dislodged during the contractile movements of larvae and/or sample fixation and washing. A similar accumulation of hemocytes was observed in eye imaginal discs (see below), as well as in discs in which cell competition was induced by generating wildtype clones in a *Minute* background (Fig. S9).

The number of hemocytes in discs with 2x *dMyc* clones was almost maximal at 16 h ACI (Fig. 2D). Interestingly, the spatial distribution of hemocytes appeared relatively uniform at 48 h, but by 72 h they exhibited a noticeable preference for being within a few microns of *dMyc* clonal boundaries (Fig. 2E).

The observation that hemocytes are recruited during cell competition would seem to contradict the findings of an earlier study[57]. To try to reconcile this difference, we carefully examined the influence of handling and staining procedures on the numbers of hemocytes

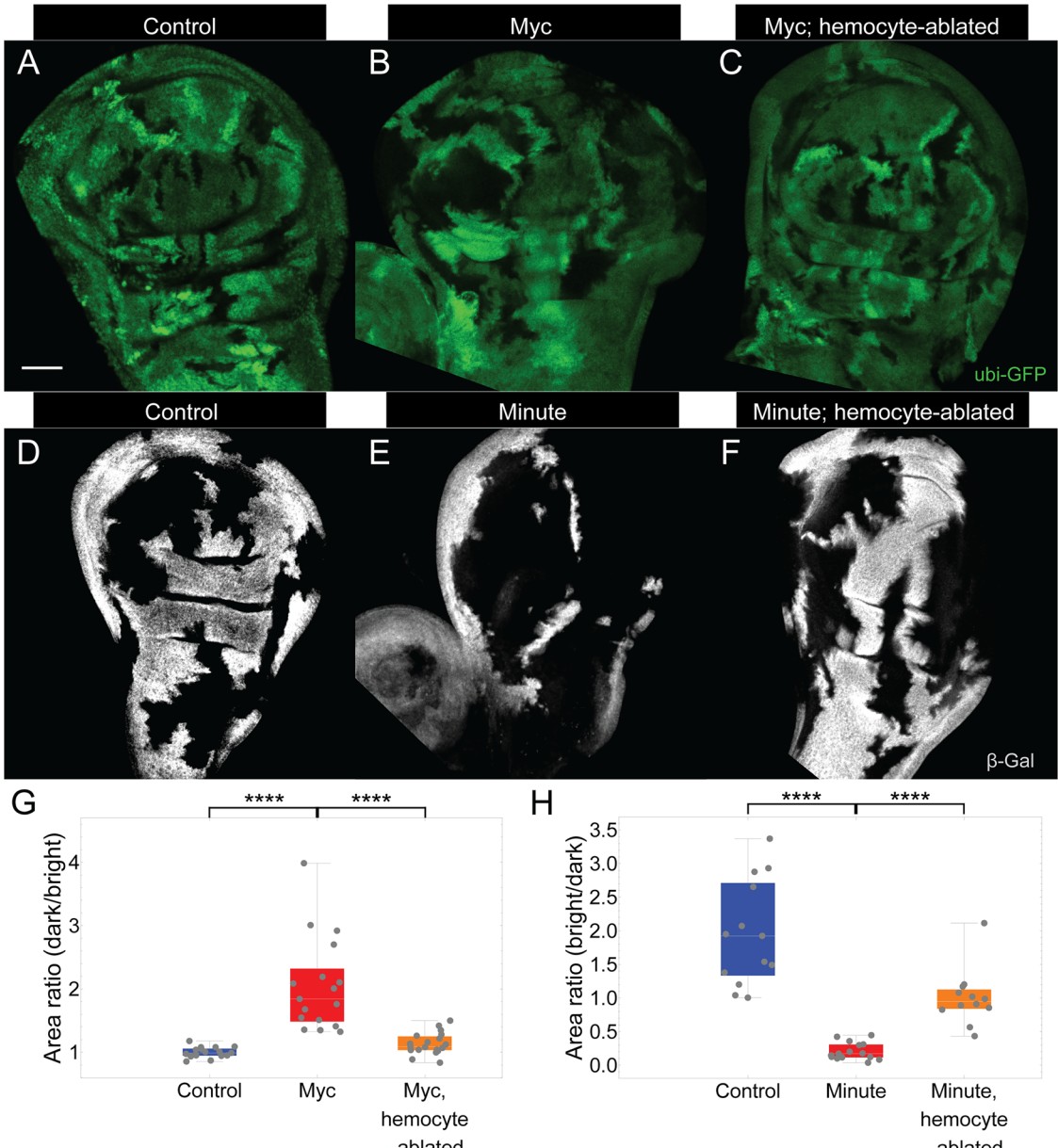

**Fig. 1 | Cell competition is hemocyte-dependent. A–F** Twin-spot mosaic clones in wing discs. Control clones (**A**) have either zero (black) or two (bright green) copies of GFP in a background carrying an extra copy of *dMyc*; *dMyc* clones (**B**) either have two extra copies of *dMyc* (black) or no extra copies of *dMyc* (bright green). In (**C**) clones were generated as in (**B**), but hemocytes were ablated using *hml-Gal4* driving *UAS-hid*. 72 h after clone induction (ACI). Control clones (**D**) have either zero (black) or two (bright gray) copies of *arm-lacZ*; *Minute* clones (**E**) have either zero (black) or two (bright gray) copies of *Minute*. In (**F**) clones were generated as in (**E**), but hemocytes were ablated using *hml-Gal4* driving *UAS-hid*. 96 h ACI.

Quantification of the area ratio between twin clones (dark/bright, or bright/dark) in wing discs of the genotypes in **A–C** (**G**) or **D–F** (**H**), respectively (****$P < 0.0001$ by two-sided Mann-Whitney U test). $P$ value = $6.45 \times 10^{-7}$, $1.35 \times 10^{-6}$, $7.08 \times 10^{-6}$, and $1.41 \times 10^{-5}$, respectively. $n = 17, 17, 19, 13, 15$, and 12 discs for the genotypes in (**A**), (**B**), (**C**), (**D**), (**E**), and (**F**), respectively. Scale bar = 50 μm. Anterior is to the left and posterior to the right. In each box and whisker plot, the box represents the interquartile, spanning from the first quartile to the third quartile, with a white line inside marking the median, and the whiskers show the range of the data, reaching the minimum and maximum values. Source data are provided as a Source Data file.

associated with wing discs. As shown in Figs. S10 and S12, widely varying numbers of hemocytes can be found attached to freshly dissected discs, but with gentle washing, most detach. This is perhaps not surprising, as imaginal discs are essentially suspended in hemocyte-containing hemolymph in vivo. After 3-4 washes, however, most hemocytes associated with wildtype discs fall away, whereas in discs displaying cell competition, greater numbers remain. With additional washing even these numbers fall, and if washing is vigorous few hemocytes remain in either type of disc (Figs. S10Q and S12F), perhaps explaining how hemocyte recruitment might have been overlooked in past.

These results are consistent with the hypothesis that hemocytes bind weakly to discs in general, but cell competition induces tighter attachment. Interestingly, studies of epithelial wounding in *Drosophila* larvae have shown that hemocytes are recruited to sites of injury primarily through increased attachment and cell spreading, rather than directed migration[70,71]. To determine whether hemocyte attachment during cell competition might also be accompanied by cell spreading, we quantified the areas of hemocyte nuclei (as visualized using the nuclear RFP marker). A marked increase in nuclear area (consistent with cell flattening) was indeed seen, not only in discs examined after moderate washing (Fig. 2F-I) but even in those examined without any

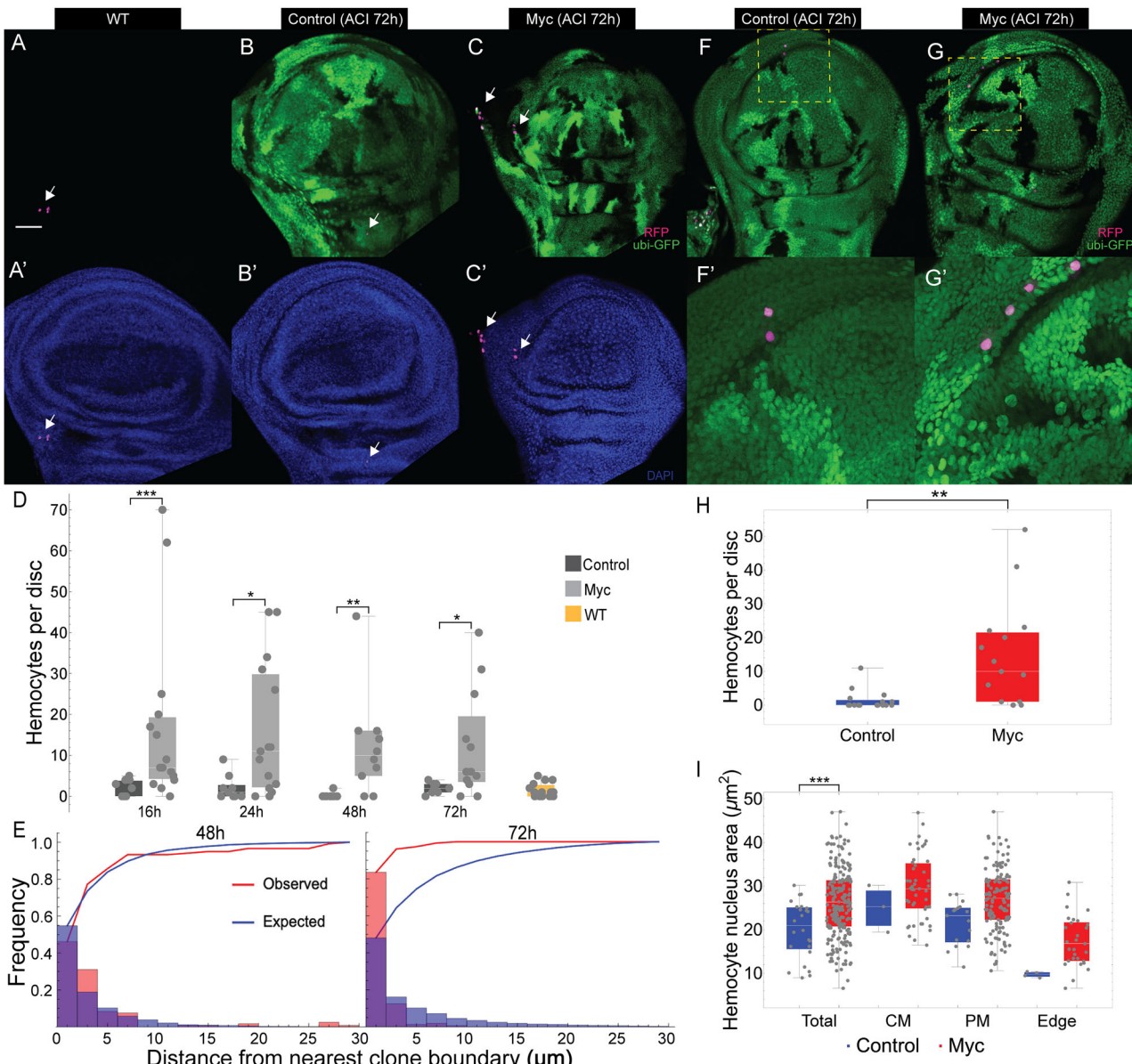

**Fig. 2 | Mosaic clones recruit hemocytes.** Wing imaginal discs carrying no (**A**), control (**B**) and *dMyc* clones (**C**), 72 h ACI; hemocytes visualized using *hml-Gal4* to drive *UAS-RFP*. (**A′-C′**) Discs in (**A-C**) were stained with DAPI. **D** Quantification of the number of hemocytes in discs of the genotypes in (**A**), or (**B**, **C**) collected at defined time periods--16, 24, 48 or 72 h ACI. *n* = 11, 15, 9, 15, 9, 10, 8, 12, and 15 discs, respectively. **E** Comparison between expected and observed distributions of hemocytes with respect to distance from clone boundaries in wing discs of the genotype in **C**, collected at 48 or 72 h ACI (See Methods). Wild-type (**F**) and *dMyc*-overexpressing (**G**) clones were induced in wing discs, and hemocytes were detected using *hml-Gal4* driving *UAS-RFP*, 72 h ACI. **F′** and **G′** are higher magnification views of the dashed areas in **F** and **G**, respectively. **H** Numbers of hemocytes in wing discs of the genotypes in **F** and **G**; *n* = 16 and 15 discs, respectively. **I** Areas of hemocyte nuclei in discs of the genotypes in (**F**) and (**G**). *n* = 24 and 215 hemocytes for the genotypes in (**F**) and (**G**), respectively. Observations were subdivided into groups based on the part of the disc examined (Columnar membrane, CM; Peripodial membrane, PM; Edge). *$P < 0.05$; **$P < 0.01$; ***$P < 0.001$ by two-sided Mann-Whitney U test. *P* value = $8.95 \times 10^{-4}$, $1.23 \times 10^{-2}$, $4.27 \times 10^{-3}$, $1.86 \times 10^{-2}$, $3.44 \times 10^{-2}$, and $3.42 \times 10^{-4}$, respectively. Bar = 50 μm. In each box and whisker plot, the box represents the interquartile, spanning from the first quartile to the third quartile, with a white line inside marking the median, and the whiskers show the range of the data, reaching the minimum and maximum values. Source data are provided as a Source Data file.

washing (Fig. S10R-V), in which two distinct pools of nuclear size could be recognized. Overall, the results suggest that discs containing competing cells recruit hemocytes primarily by promoting adhesion and spreading on epithelial basal surfaces or their associated basement membranes.

## Hemocyte recruitment occurs regardless of whether cells die or competition succeeds

An obvious reason why hemocytes might be recruited to sites of cell competition is that loser cells undergo apoptosis there, and

hemocytes—like their vertebrate counterparts, macrophages—home to sites of cells death, where they carry out the phagocytic removal of cell corpses. Indeed, the death and removal of loser cells is necessary for cell competition to proceed successfully[51,65]. Based on such results, one might expect that any role of hemocytes in cell competition would be connected to their ability to phagocytose dead cells. Indeed, others have suggested as much[29,72].

To test whether dying cells are actually what recruit hemocytes, we generated 2x *dMyc* twin-spot clones in backgrounds in which cell death was blocked. In one case, we generated clones on a *hid*/+ mutant

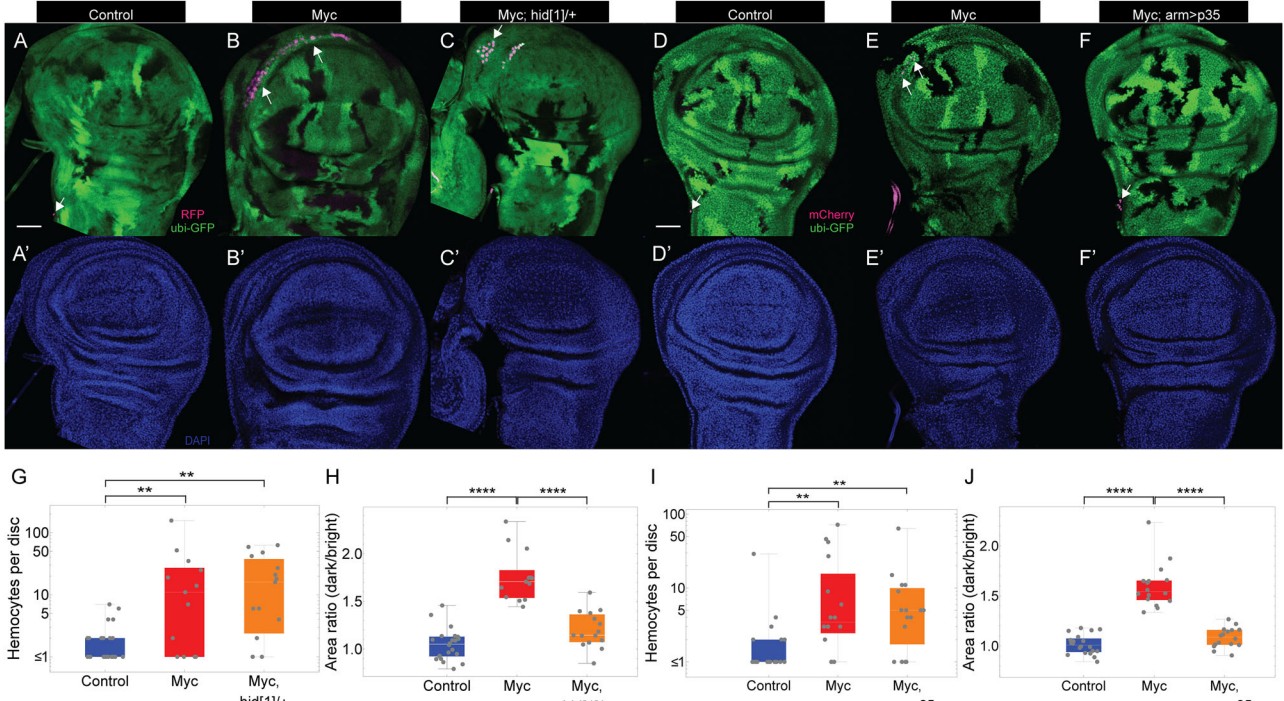

**Fig. 3 | Hemocyte recruitment when apoptosis is blocked. A–F** Control and Myc clones were generated and hemocytes visualized as in Fig. 2F, G, 72 h ACI. In (**C, F**) clones were generated as in (**B, E**), but cell death was blocked using a copy of *hid* allele (*hid[1]*) (**C**) or *arm-Gal4* driving *p35* (**F**). (**A′–F′**) Discs in (**A–F**) were stained with DAPI. Scale bar = 50 μm. Quantification of the number of hemocytes in discs of the genotypes in **A–C** (**G**) or **D–F** (**I**). (**H, J**) Quantification of the area ratio between twin clones (dark/bright) in discs of the genotypes in **A–C** (**H**) or **D–F** (**J**). \*\*P < 0.01;

\*\*\*\*P < 0.0001, by two-sided Mann-Whitney U test. P value = $6.65 \times 10^{-3}$, $1.78 \times 10^{-3}$, $1.25 \times 10^{-6}$, $1.65 \times 10^{-5}$, $1.23 \times 10^{-3}$, $2.91 \times 10^{-3}$, $3.51 \times 10^{-7}$, and $1.41 \times 10^{-6}$, respectively. $n = 22$, 13, 15, 20, 16, and 16 discs for the genotypes in (**A**), (**B**), (**C**), (**D**), (**E**), and (**F**), respectively. In each box and whisker plot, the box represents the interquartile, spanning from the first quartile to the third quartile, with a white line inside marking the median, and the whiskers show the range of the data, reaching the minimum and maximum values. Source data are provided as a Source Data file.

background (*hid[1]*, an antimorphic allele[73]), which produces a strong block of apoptosis throughout the larva; in the other the apoptosis inhibitor p35 was expressed under the control of *arm-Gal4*. In *hid[1]/+* wing discs, cell competition was blocked, but hemocyte recruitment was not significantly different from that observed in wing discs that did not have a copy of *hid[1]* (Fig. 3A–C, G–H). Similar results were observed in *hid[1]/+* eye discs bearing *dMyc* twin-spot clones (Fig. S11). Similarly, in *arm-Gal4 > p35* wing discs, cell competition was blocked and hemocyte recruitment was unaffected (Fig. 3D–F, I–J). These results establish that signals independent of those produced by dying cells are responsible for hemocyte recruitment.

## Hemocytes are recruited by clonal juxtapositions, not by *dMyc* expression itself

To determine whether the recruitment of hemocytes is driven by the expression of *dMyc* itself, as opposed to the cell competition that *dMyc* induced, we attempted to induce 2x *dMyc* as uniformly as possible throughout the entire wing disc. To this end, we combined the FRT-FLP technique with the *Gal4/UAS* system to create GFP-labelled flip-out clones that carry an extra copy of *dMyc* in *Drosophila* wing imaginal discs, and tracked hemocytes using *srpHemo-H2A-3xmCherry* (using either of two versions of the transgene, one on the second and one on the third chromosome)[69]. Flip-out *dMyc* clones displayed cell competition similar to that observed with twin-spot clones (Fig. S14A–G), as expected[29,51,55,57,67]; they also displayed cell death that was disproportionally localized to wild type cells surrounding the *dMyc*-expressing clones (Fig. S14H–K).

In one set of larvae, we heat-shocked for 10 min to create dMyc clones sporadically throughout wing discs, and harvested 14 h after clone induction (Figs. 4B and S15B). In another of the same genotype, we heat-shocked for 1 h, to induce *dMyc* expression in as many cells as

possible (Figs. 4C and S15C). As a control, wild-type clones were created with a 10-min heat-shock in a wild-type background (Figs. 4A and S15A).

Consistent with our results using twin-spot clones, there was a large increase in the number of disc-associated hemocytes in discs containing *dMyc* clones. In contrast, when *dMyc* was expressed in near-uniform fashion, we saw no increase in hemocyte numbers over that in discs with wild-type clones (Figs. 4G and S15D). Likewise, when we generated wing discs with uniform expression of two extra copies of *dMyc* (Fig. 4F and F′) hemocyte recruitment was not observed (Fig. 4D–F′ and H). These results demonstrate that it is not *dMyc* overexpression per se, but rather the juxtaposition of cells with different levels of *dMyc* that drives hemocyte recruitment.

## Expression of *eiger/TNF* by hemocytes is necessary for cell competition

Neither the mechanism by which winner and loser cells are distinguished during cell competition, nor the mechanism by which loser cells are eliminated, is fully understood, but several signaling pathways have been implicated in these processes. For example, during cell competition, Jun N-terminal kinase (JNK) signaling is elevated in both winner and loser cells, and is at least partially required for apoptosis[50,51,74]. Moreover, evidence indicates that the single JNK protein in *Drosophila*, encoded by *basket* (*bsk*), is required for cell competition[45] (although one recent study[58] has questioned its essentiality). Consistent with a central role for JNK signaling in cell competition, we observed that hemocyte ablation also blocked JNK activation in wing discs undergoing *dMyc*-dependent cell competition (Fig. S16).

The secreted TNF ortholog Eiger potently activates JNK signaling in *Drosophila* (just as TNF is a potent JNK agonist in mammals), and in flies mutant for *eiger*, cell competition is abrogated[58]. The cellular

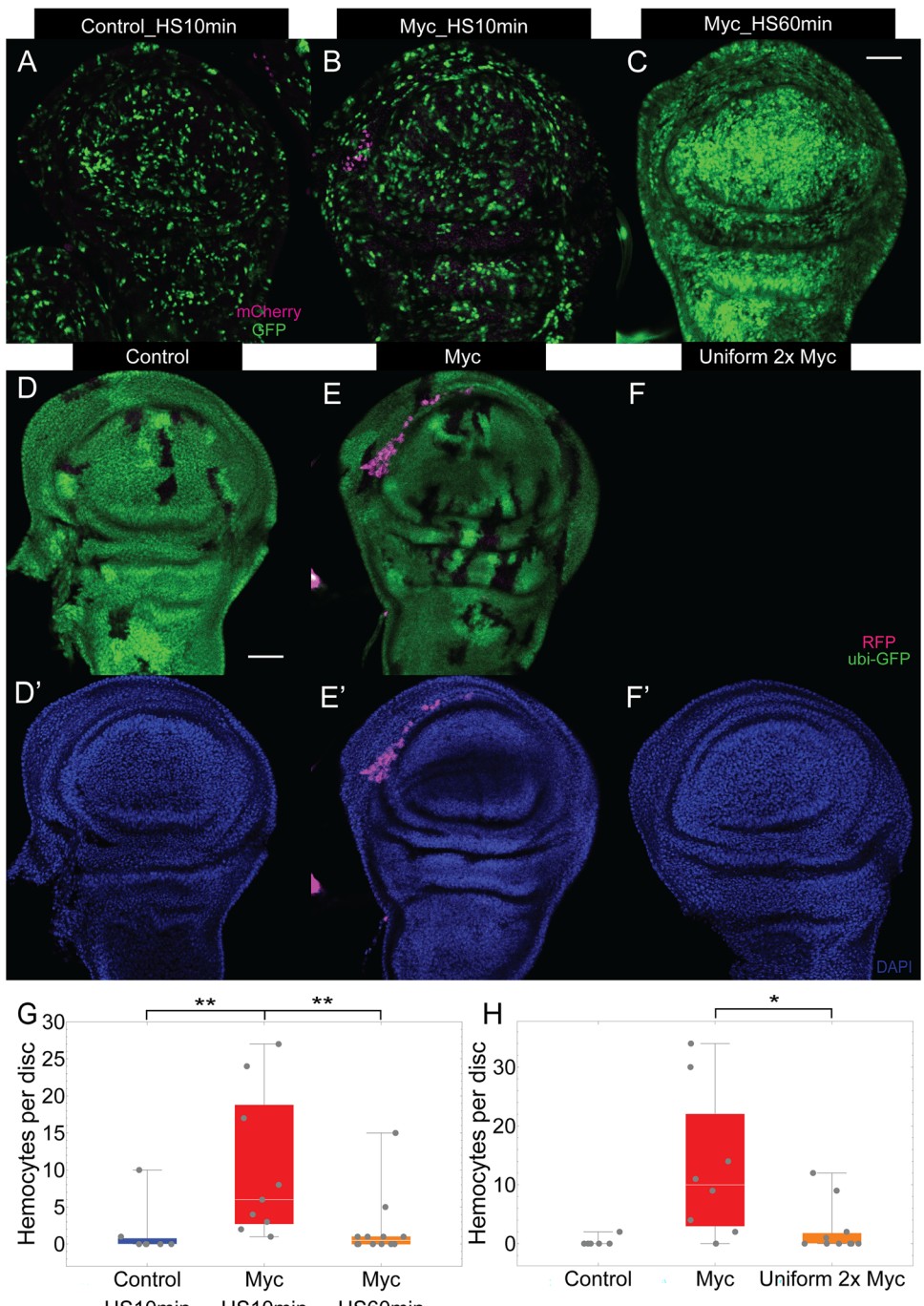

**Fig. 4 | Hemocytes are recruited by cell juxtapositions.** Flip-out mosaic clones in wing discs at 14 h ACI. Wild-type (**A**) and *dMyc*-overexpressing (**B**, **C**) clones were created by 10-min (**A**, **B**) or 1-hr (**C**) heat shock. Hemocytes were visualized by *srpHemo-H2A-3xmCherry* (III) (**A**–**C**). Wing discs carrying control clones (**D**), *dMyc* clones (**E**) and a nearly uniform field of cells expressing two copies of *dMyc* (**F**), 72 h ACI; hemocytes visualized using *hml-Gal4* to drive *UAS-RFP*. (**D'**–**F'**) Wing discs in (**D**–**F**) were stained with DAPI. Numbers of hemocytes in wing discs of the genotypes in **A**–**C** (**G**) or **D**–**F** (**H**). *$P < 0.05$; **$P < 0.01$, by two-sided Mann-Whitney U test. *P* value = $8.14 \times 10^{-3}$, $3.19 \times 10^{-3}$, and $1.49 \times 10^{-2}$, respectively. $n = 7, 9, 12, 6, 8$, and 11 discs for the genotypes in (**A**), (**B**), (**C**), (**D**), (**E**), and (**F**), respectively. Scale bar, 50 μm. In each box and whisker plot, the box represents the interquartile, spanning from the first quartile to the third quartile, with a white line inside marking the median, and the whiskers show the range of the data, reaching the minimum and maximum values. Source data are provided as a Source Data file.

source of Eiger during cell competition is currently unclear. Disc epithelial cells express Eiger at very low levels, and a significant increase in expression is not seen in discs undergoing Myc-induced cell competition[58]. Nevertheless, disc-derived Eiger has been implicated in tumor suppression[24], as well as Eiger derived from other sources. For example, *eiger* is highly expressed by stimulated hemocytes[25,75] (just as TNF is a major secreted product of activated macrophages) and hemocyte *eiger* has been implicated in *Drosophila* tumor

suppression[25], although recent evidence suggests that the fat body, which releases Eiger into the circulation, may be the primary source of Eiger during tumor suppression[32].

Given the data implying an important role for hemocytes in cell competition, we wondered whether this role required hemocyte-derived Eiger. To test this hypothesis, we carried out hemocyte-specific RNAi-mediated knockdown, using *hml-Gal4* to drive either of two *UAS-egr-RNAi* constructs[76–80], in some cases

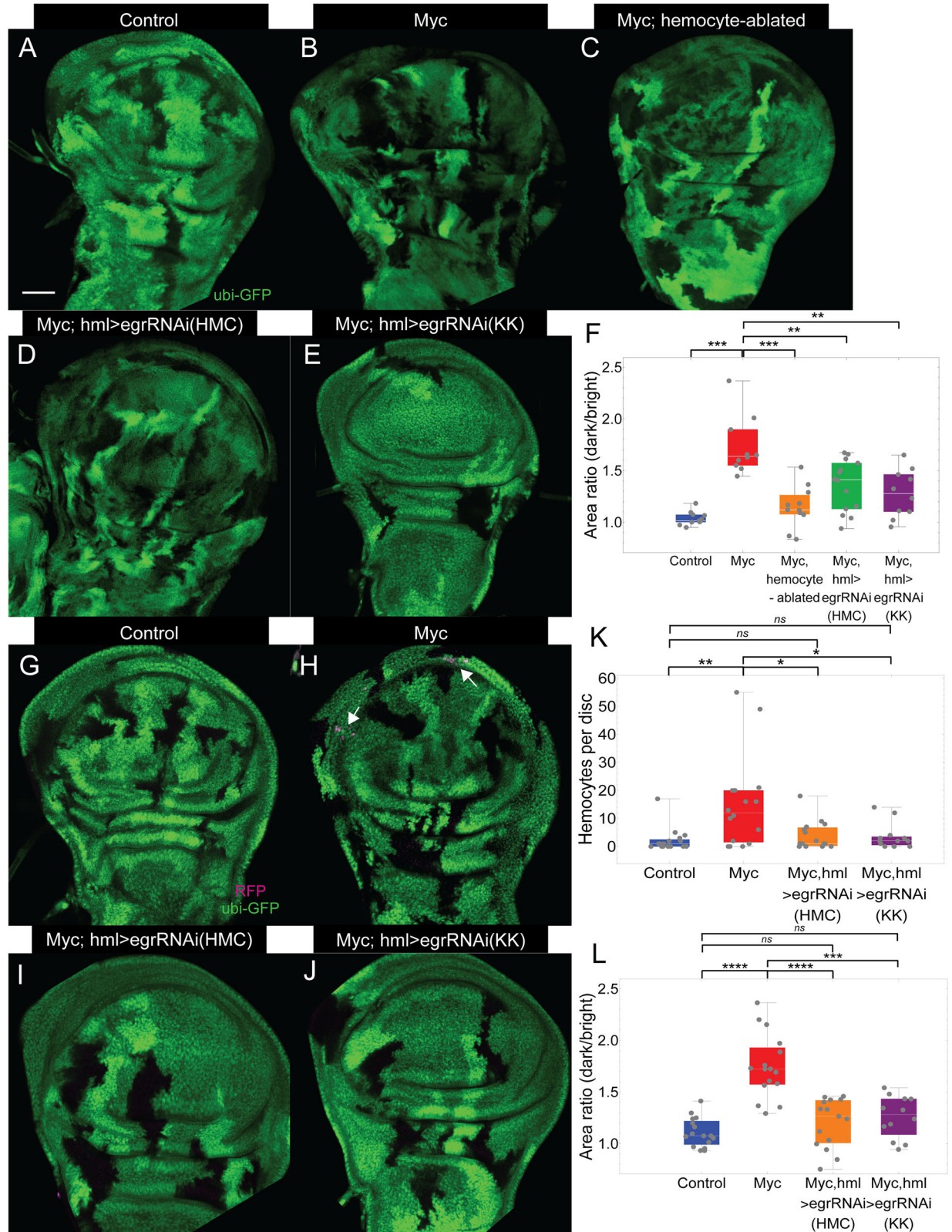

together with *UAS-RFP*. Both cell competition and hemocyte recruitment were assessed (Fig. 5). Intriguingly, when *egr* was knocked down just in hemocytes, the area ratio between 2x *dMyc* and wild-type cells was markedly reduced, although not as completely as when hemocytes were ablated (Fig. 5A–F). Notwithstanding the possibility that RNAi-mediated knockdown may not be complete, the results indicate that a large part of the

contribution of hemocytes to cell competition depends on their production of Eiger.

Whereas *dMyc* twin-spot clones offer advantages for the precise quantification of cell competition (2x *dMyc* and wild-type areas should be equal in the absence of successful competition), the presence of three types of clonal boundaries (2X-1X; 2X-wildtype; and 1X-wildtype) potentially complicates the interpretation of underlying mechanisms.

**Fig. 5 | Role of *Eiger* in cell competition.** In (**A**–**C**) Clones were generated as in Fig. 1, whereas in (**D**, **E**) clones were generated as in (**B**), but *egr* was knocked down in hemocytes, using *egrRNAi* (HMC/KK). **F** Quantification of the area ratio between twin clones (dark/bright) in discs of the genotypes in **A**–**E**. In **G**–**J**, clones were induced and hemocytes visualized as in Fig. 3. In **I** and **J** discs were manipulated as in (**H**), but *egr* was knocked down in hemocytes as in **D** and **E**, respectively. **K** Quantification of the number of hemocytes in wing discs of the genotypes in **G**–**J**. **L** Quantification of the area ratio between twin clones (dark/bright) in discs of the genotypes in **G**–**J**. *$P < 0.05$; **$P < 0.01$; ***$P < 0.001$; ****$P < 0.0001$ by two-sided

Mann-Whitney U test. *P* value = $1.08 \times 10^{-4}$, $1.90 \times 10^{-4}$, $5.92 \times 10^{-3}$, $1.94 \times 10^{-3}$, $6.66 \times 10^{-3}$, $2.44 \times 10^{-1}$, $2.55 \times 10^{-1}$, $2.69 \times 10^{-2}$, $4.85 \times 10^{-2}$, $2.96 \times 10^{-6}$, $1.44 \times 10^{-1}$, $5.12 \times 10^{-2}$, $4.67 \times 10^{-5}$, and $1.41 \times 10^{-4}$, respectively. $n = 11, 10, 11, 14, 10, 16, 16, 15$, and 12 discs for the genotypes in (**A**), (**B**), (**C**), (**D**), (**E**), (**G**), (**H**), (**I**), and (**J**), respectively. Bar = 50 μm. In each box and whisker plot, the box represents the interquartile, spanning from the first quartile to the third quartile, with a white line inside marking the median, and the whiskers show the range of the data, reaching the minimum and maximum values. Source data are provided as a Source Data file.

Therefore, we also carried out these experiments under conditions in which wild-type flip-out clones are induced in a *dMyc*-expressing background. Such wild-type clones are efficiently eliminated by cell competition; indeed, to observe large numbers of them one needs to increase the duration of heat shock compared to the amount required for control clones (Fig. S17A-E). When hemocytes were labeled with *srpHemo-H2A-3xmCherry*, we not only observed that such cells were recruited both by wild-type clones in a *dMyc* background and *dMyc* clones in a wild-type background, but also that the number of recruited hemocytes increased with the number of clones per disc (Fig. S17F, G).

Furthermore, when hemocytes were ablated, or hemocyte-derived Eiger was blocked, cell competition in this setting was strongly inhibited. In one set of experiments, in which the LexA system was used to generate clones and *hml*-Gal4 to either eliminate hemocytes or knockdown hemocyte Eiger, both treatments restored clones to a size not significantly different from wildtype clones in a wildtype background (Fig. S17H–L). In another set of experiments, in which the Gal4 system was used both to generate clones and express RNAi, knocking down Eiger in both hemocytes and loser clones markedly increased clone sizes (Fig. S17M-P). Interestingly, knocking down Eiger in loser clones alone also had a small but statistically significant effect, implying that epithelial Eiger does contribute, albeit modestly, to cell competition. These findings indicate that, regardless of the methods used to generate *dMyc*/wild-type clonal boundaries, the ensuing cell competition is associated with hemocyte recruitment, and depends on hemocyte Eiger.

An unexpected finding was that hemocyte-specific knockdown of Eiger significantly reduced hemocyte recruitment to discs undergoing cell competition (Fig. 5G–L). This result suggests that hemocyte-derived Eiger either acts on hemocytes themselves or boosts the ability of epithelial cells to recruit hemocytes (see Discussion).

### Recruitment of hemocytes does not require JNK signaling
To test whether JNK signaling plays a role in hemocyte recruitment, we generated *dMyc* twin spot clones in a variety of genetic backgrounds in which JNK signaling was inhibited, either throughout the larva, or specifically in hemocytes (Fig. 6).

As expected, when JNK was eliminated using a *bsk* loss-of-function allele, cell competition was blocked either partially (*bsk*/+) or completely (*bsk*/*bsk*) (Figs. 6B and S18). Cell competition was also blocked when a JNK dominant negative allele (*bsk*[DN])[81,82] was driven by the broadly expressed *arm-Gal4* driver; Fig. 6D). In contrast, hemocyte recruitment was not affected in *bsk*/+ wing discs, nor was it significantly reduced in wing discs in which *arm-Gal4* drove *bsk*[DN]. Recruitment was partially decreased in *bsk*/*bsk* wing discs (Fig. 6A, C), but given that few homozygous *bsk* flies survive, it is possible that reduction of hemocyte recruitment in this case was a reflection of poor larval health. These results suggest that hemocyte recruitment is not only independent of cell death, but also relatively independent of JNK signaling (suggesting that the above-mentioned requirement for Eiger in hemocyte recruitment may be mediated by JNK-independent Eiger signaling, e.g.[58,83]).

In parallel experiments, we used *hml-Gal4*, driving either *bsk*[DN], or *bsk*-RNAi (the activity of which we validated; Fig. S19), to knock down

JNK specifically in hemocytes (Fig. 6A, B). Consistent with the results of widespread *bsk* knockdown, we did not observe a significant reduction in hemocyte recruitment to sites of cell competition (Fig. 6A). Surprisingly, however, we did see a strong, albeit incomplete, inhibition of cell competition (Fig. 6B). These results suggest that, while JNK signaling is not required for hemocyte recruitment, it is needed in both epithelial cells and hemocytes for effective cell competition to occur.

### Hemocytes contribute to killing loser cells
Given the evidence that Eiger production by hemocytes is required for cell competition (Fig. 5), we investigated whether hemocytes contribute to the killing of loser cells. To this end, we labelled dying cells using anti-Dcp1 antibodies in discs in which *dMyc*-induced cell competition was generated, and hemocytes ablated, as in Fig. 1, with fixation at two different times (16 or 72[25] h ACI). As shown in Fig. 7(A-C", E-F"" and H-I), the presence of *dMyc* twin spot clones led to a marked elevation of Dcp1 staining within 16 h after clone induction, as expected.

Levels of Dcp1-staining should reflect not only on the rate at which cells die but also the rate at which dead cells are removed. Since the latter process depends largely on hemocytes, under the model that the sole role of hemocytes is a phagocytic one, one should expect hemocyte ablation to increase Dcp1 staining. In fact, the observed effect was the opposite. In hemocyte-ablated discs, the number of Dcp1-positive cells fell, although not completely to the control levels (Fig. 7C, D, F, G and H, I). These results argue that hemocytes play a role in cell killing during cell competition.

We next investigated whether hemocyte-derived Eiger is required for cell killing, As shown in Fig. S20, hemocyte-specific knockdown of Eiger significantly reduced (albeit incompletely) the increased number of Dcp1-positive cells in wing discs undergoing *dMyc*-induced cell competition. Similar results were obtained when we blocked JNK signaling (using *bsk* null, dominant negative or RNAi alleles), either in the entire larva (Fig. S21A–C, F) or just in hemocytes (S21A, B, D–F). These results imply that hemocytes are at least partially responsible for triggering apoptosis during *dMyc*-induced cell competition, and that this process requires both Eiger and JNK signaling in both epithelial cells and hemocytes.

Finally, we asked whether similar results could be obtained for *Minute*-induced cell competition. As with *dMyc*, *Minute* induced cell competition was accompanied by elevated Dcp1 staining, and either hemocyte ablation or hemocyte-specific knockdown of *egr* led to a statistically significant decrease in staining (Fig. S22A'–F', G'-L', N, P), although the effect of hemocyte ablation on *Minute* cell competition is not as complete as it is for *dMyc* cell competition (Figs. 1 and S22M, O).

Together, these results suggest that hemocytes, via secretion of Eiger, contribute to the killing of loser cells during cell competition.

### Discussion
Terms such as "cell competition," "interface surveillance", "tumor suppression", and "tumor surveillance" have been applied to diverse phenomena, in vertebrates and invertebrates, in which cells that differ from their neighbors are eliminated[59–63,84]. In tumor suppression[25] and tumor surveillance[4–13], immune cells are thought to play an active role–consistent with the ability of the immune system to attack established

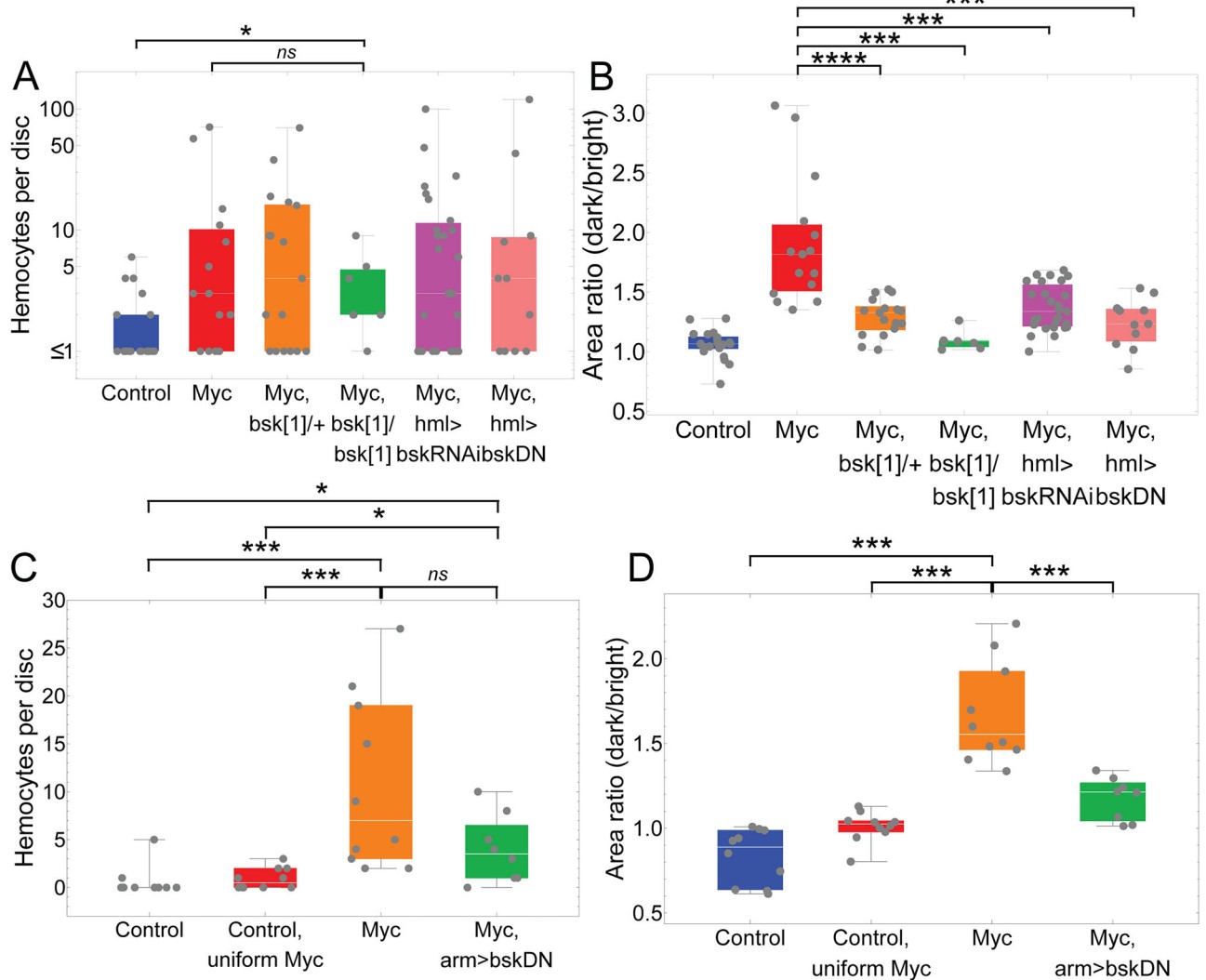

**Fig. 6 | Role of JNK in cell competition.** Hemocyte recruitment (**A**, **C**) and cell competition (**B**, **D**) in wing discs in which JNK signaling was manipulated. **A, B** Control and Myc clones were generated and hemocytes visualized as in Fig. 3, 72 h ACI; clones in Myc,bsk[1]/+ and Myc,bsk[1]/bsk[1] were created in wing discs carrying a heterozygous or homozygous *bsk* loss-of-function mutation (*bsk¹*); In Myc,hml>bskRNAi and Myc,hml>bskDN discs *bsk* was specifically knocked down in hemocytes using *bskRNAi* or *bskDN*. *n* = 21, 15, 17, 7, 27, and 11 for the genotypes in (**A**), respectively. **C, D** clones were created as in **A, B**, but hemocytes were visualized by *srpHemo-H2A-3xmCherry*; clones denoted as "Control uniform Myc" were generated as in Fig. 1A; In clones marked Myc,arm>bskDN, *bsk* was knocked down

everywhere using *arm-Gal4* to drive *UAS-bsk^DN*. *n* = 10, 10, 10, and 8 discs for the genotypes in (**C**), respectively. *\*P* < 0.05; \*\*\**P* < 0.001; \*\*\*\**P* < 0.0001 by two-sided Mann-Whitney U test. *P* value = $1.01 \times 10^{-2}$, $1.00 \times 10^{0}$, $3.01 \times 10^{-5}$, $2.15 \times 10^{-4}$, $1.57 \times 10^{-4}$, $1.36 \times 10^{-4}$, $5.83 \times 10^{-4}$, $1.13 \times 10^{-2}$, $5.83 \times 10^{-4}$, $3.30 \times 10^{-2}$, $1.00 \times 10^{-1}$, $1.57 \times 10^{-4}$, $1.57 \times 10^{-4}$, and $5.30 \times 10^{-4}$, respectively. In each box and whisker plot, the box represents the interquartile, spanning from the first quartile to the third quartile, with a white line inside marking the median, and the whiskers show the range of the data, reaching the minimum and maximum values. Source data are provided as a Source Data file.

cancers[1–3]—but it has been argued that the function of immune cells in instances of cell competition not involving tumor cells is limited to the removal of cell corpses[28,29,59]. Here we challenge this view, by showing that, in *Drosophila*, the innate immune cells known as hemocytes are required for and recruited during classical examples of cell competition; their recruitment is independent of cell killing and JNK signaling; and both the cell killing associated with cell competition, and the success of cell competition itself, depend upon the production of the TNF ligand Eiger, specifically by hemocytes.

If tumor suppression shares so many features with other types of cell competition, one might question why, if in the latter case losers tend to be cells that are expected to be at a fitness disadvantage, e.g. due to slower growth, how is it that in the former case cells with the propensity to grow uncontrollably get eliminated? As others have pointed out, even though slow growth sometimes correlates with loser status, growth rate appears not to be the determining factor in cell

competition. Not all juxtapositions of cells that grow at different rates lead to cell competition, and not all tumorigenic genotypes get eliminated[59,60]. Moreover, changes in insulin/mTor signaling pathways in tumor cells have been reported to turn cells that are destined to lose into ones that reliably overgrow[85]. Recently it has been suggested that the common trigger in all cases has less to do with differential fitness than with the existence of stress—mechanical, proteostatic, metabolic, oxidative, or some combination of these[86–89]—a view that could help explain how so diverse an array of cellular juxtapositions can converge on a common process.

The results presented here support the view that Eiger and JNK signaling are essential in both tumor-suppressive and non-tumor-suppressive types of cell competition and provide evidence that hemocytes are an important source of Eiger in both cases. Yet hemocytes are unlikely to be the only relevant source. Indeed, studies of tumor suppression have produced evidence in favor not only of

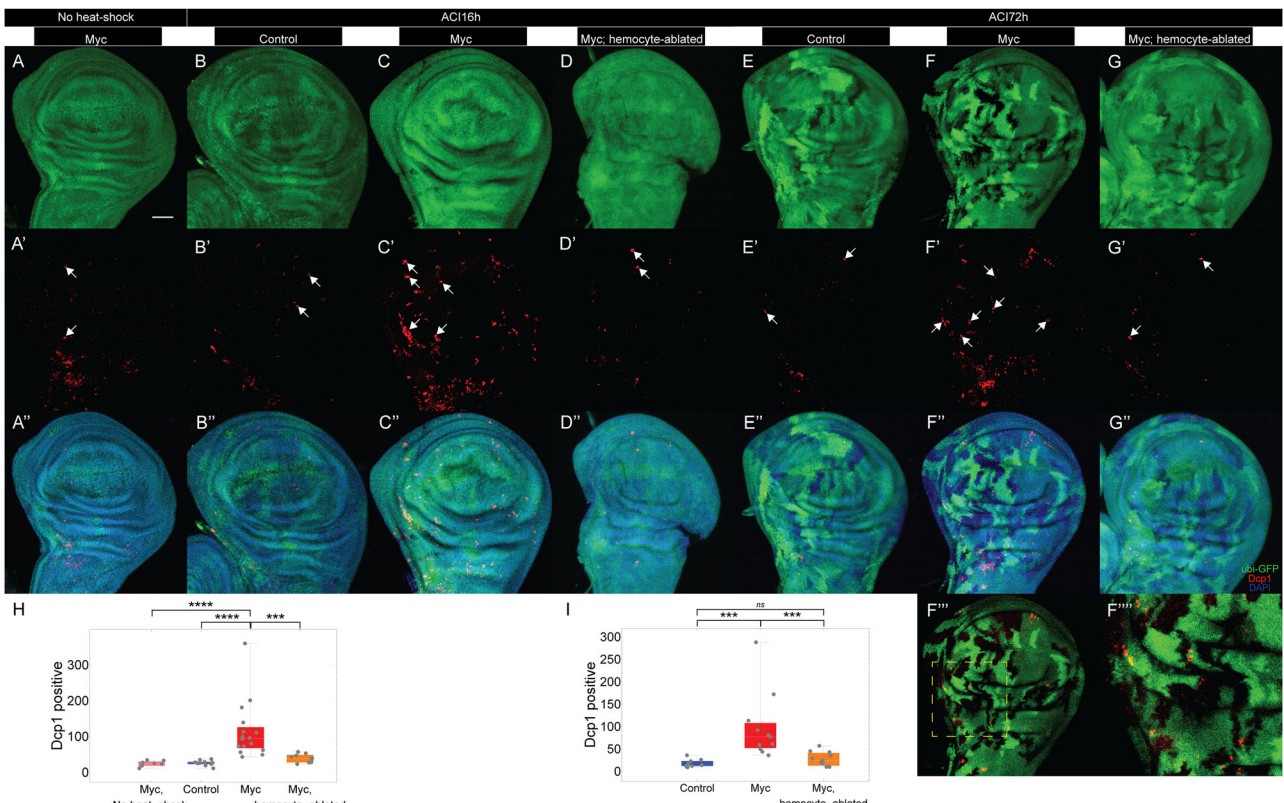

**Fig. 7 | Hemocytes contribute to triggering apoptosis in loser cells.** Cell competition was induced in wing discs using *dMyc* twinspot clones, with or without hemocyte ablation, as in Fig. 1. Wing discs were imaged for GFP fluorescence (**A-G**), anti-Dcp1 immunostaining (red, **A'-G'**), and DAPI (light blue, with all three channels overlaid in **A"-G"**). **A** No heat-shock control (genotype as in Fig. 1B); **B-D** panels show control clones (**B**), *dMyc* clones (**C**) and *dMyc* clones plus hemocyte ablation (**D**), 16 h ACI. **E-G** panels show control clones (**E**), *dMyc* clones (**F**) and *dMyc* clones plus hemocyte ablation (**G**), 72 h ACI. All images are maximum intensity projections, except for **F'''**, in which a single optical slice of the disc in F is shown, along with a higher magnification view of a portion of that image (**F''''**). **H** Dcp1-positive cells per disc for the genotypes in **A-D**. **I** Dcp1-positive cells per disc for the genotypes in **E-G**. In **A'-G'**, representative Dcp1-positive cells are indicated with white arrows. Data in **H** and **I** include all Dcp1-positive cells, both apical and basal (***$P < 0.001$; ****$P < 0.0001$ by two sided Mann-Whitney U test). $P$ value = $8.88 \times 10^{-5}$, $2.48 \times 10^{-5}$, $2.09 \times 10^{-4}$, $1.24 \times 10^{-4}$, $6.20 \times 10^{-2}$, and $3.45 \times 10^{-4}$, respectively. $n = 8, 10, 16, 9, 10, 11$, and $11$ discs for the genotypes in (**A**), (**B**), (**C**), (**D**), (**E**), (**F**), and (**G**), respectively. Bar = 50 μm. In each box and whisker plot, the box represents the interquartile, spanning from the first quartile to the third quartile, with a white line inside marking the median, and the whiskers show the range of the data, reaching the minimum and maximum values. Source data are provided as a Source Data file.

hemocyte-derived Eiger[25], but also epithelial cell-derived Eiger[24] and systemic Eiger (derived from the fat body)[32] as each being required. The results obtained here support the view that, during cell competition, Eiger derives from at least two sources: hemocytes and, to a lesser extent, loser cells themselves (Fig. S17).

We speculate that what seems like a puzzling degree of redundancy may reflect the unique needs of a system that must turn small differences between neighboring cells into a binary decision to eliminate one cell and not the other. Because factors like Eiger, and the JNK signal it triggers, can cause the death of cells in general, successful cell competition requires a large difference in some type of signal between adjacent cells—even ones that differ only subtly in phenotype—such that one definitively wins and the other definitively loses.

Such decision-making is unlikely to be explained by the distributions of diffusible extracellular factors (such as Eiger), as winner cells will presumably be exposed to almost the same levels of such molecules as their losing neighbors. Processes downstream must somehow generate a switch-like response within responding cells, so that losers fall above a threshold for killing and winners below. The most common way signaling systems do this is through combinations of positive and negative feedback loops that produce bi-stability[90,91]. It is noteworthy that both kinds of feedback loops exist in the JNK pathway of wing disc cells: JNK stimulates itself by upregulating production of reactive oxygen species (ROS) which in turn increase JNK activity[62,92,93] while at the same time inhibiting itself through the upregulation of *puckered*[94].

One of the drawbacks of bi-stable switches, however, is they are susceptible to noise, i.e. they may flip at random when close to the switching threshold. Unwanted transitions can be avoided by raising the threshold, but then desired ones may fail to occur. One way around this problem is to maintain a high threshold under most circumstances, and then lower it when true signals are expected. This is what we imagine may be occurring in wing discs: Epithelial cells produce Eiger, but at a level insufficient to trigger sustained JNK signaling in most loser cells. Upon recruitment of hemocytes—or even upon the systemic release of Eiger[32]—levels of Eiger would rise enough to enable loser cells to cross the switching threshold and commit to die.

While speculative, this model agrees with several observations. First, Eiger expression by imaginal disc epithelial cells is indeed very low[75], even during cell competition, often near the limits of detection[58]. Second, while hemocytes are strongly recruited to discs by cell competition, they don't initially localize to the precise sites of clonal juxtapositions (Fig. 2E), suggesting they may be less involved in determining where cells are killed than in adjusting something globally in the disc, e.g., the overall threshold for killing. Third, the reduction in cell death during cell competition that occurs upon hemocyte ablation is only partial (Fig. 7), consistent with view that epithelial cells can kill their neighbors on their own, but less efficiently.

Although the present work demonstrates recruitment of hemocytes to sites of cell competition, it does not address the nature of what recruits them, except to show that dead cells are not required (Fig. 3). Other studies in wing discs have suggested that diffusible fragments of tyrosyl-tRNA synthase[67], ROS[75,95], and the growth factor Pvf1[96] may function to promote hemocyte recruitment to imaginal discs. Interestingly, Pvf1 has also been implicated in the recruitment of hemocytes to epithelial wounds[70], a phenomenon characterized by hemocyte attachment and spreading similar to that observed in the present study[71]; moreover, hemocytes recruited to wounds produce ROS themselves[97], potentially creating positive feedback in hemocyte recruitment. Such feedback could explain our observation that hemocyte recruitment itself requires hemocyte Eiger (Fig. 5K) provided that Eiger induces ROS production by hemocytes. In that case, however, Eiger might have to do so via a JNK-independent pathway (as described in embryonic hemocytes[83]), since we did not observe a substantial requirement for hemocyte JNK in hemocyte recruitment (Fig. 6A). Interestingly, autocrine effects of TNF have also been observed in vertebrate macrophages[98,99].

In future, it will be useful to extend this work to other cases of cell competition, besides clonal juxtaposition of *Myc* and *Minute* in *Drosophila*, especially scenarios in vertebrates. Vertebrate innate immune cells—especially macrophages and NK cells—have been implicated not just in tumor-environment interaction but in tumor surveillance[4–13], and tumor surveillance has been shown to depend on immune-derived TNF-family ligands[100–102]. Although a role for such cells in vertebrate cell competition has, to our knowledge, not been investigated, other cases exist in which macrophages enable tissues to make local, binary discriminations. For example, macrophages selectively eliminate herpes virus-infected cells from mammalian cell and explant-cultures through a mechanism that requires macrophage-derived TNF, is mimicked by global application of TNF, and works not because TNF kills cells directly but because it enhances local or cell-intrinsic events that lead to cell death[103]. The suggestion that macrophages tune cell-intrinsic events so as to improve discriminations among cells in different states strongly resembles the model for cell competition we propose above. It may be that a common, potentially ancestral role of the innate immune system is to detect and respond to tissue imbalances of all kinds—differential cell stresses, uneven growth rates, viral infection status, or malignancy—by heightening the sensitivity of tissues to their own, intrinsic defense mechanisms.

## Methods
### Drosophila stocks and genetics
*Drosophila melanogaster* flies were grown in medium composed of the following ingredients per liter: 8.4 g agar; 63.05 g dextrose; 34.225 g sucrose; 9.6 g potassium sodium tartrate; 0.735 g calcium chloride; 76 g corn meal; 32 g yeast; 4 ml propionic acid; 1 g tegosept; 10 ml 95% ethanol; 0.4 ml food coloring. Vials were kept in an incubator with a 12-hour light/dark cycle at 25 °C except for time periods for heat-shock.

The following flies were used in this study: *arm-lacZ,M(2)Z,FRT40A* (a generous gift from Jose de Celis), *FRT82B,tub-HA:Myc^{w+}* (a generous gift from Peter Gallant), *y,w,hsFLP;; Act > y>Gal4,UAS-GFP.nls* (X; III) (a generous gift from Hiroshi Nakato), *puc-LacZ* (a generous gift from Nicholas E. Baker), *tub > CD2>LexA*, *tub>Myc>LexA*, and *LexOP-GFP* (generous gifts from Laura A. Johnston), *Act5C-Gal4* (BDSC_3954), *arm-lacZ,FRT40A* (BDSC_7371), *arm-Gal4* (BDSC_1560), *bsk^1* (BDSC_3088), *FRT40A* (BDSC_8212), *FRT82B* (BDSC_86313), *FRT82B,ubi-GFP* (BDSC_5188), *FRT82B,ubi-GFP* (BDSC_32655), *hid^1*(BDSC_631), *hml-Gal4(II)* (BDSC_30139), *hml-Gal4(III)* (BDSC_30141), *hsFLP,tub>Myc>-Gal4* (BDSC_64767), *srpHemo-3XmCherry(II)* (BDSC_78358), *srpHemo-3XmCherry(III)* (BDSC_78359), *srpHemo-H2A.3XmCherry(II)* (BDSC_78361), *srpHemo-H2A.3XmCherry(III)* (BDSC_78360), *UAS-bsk^{DN}(X)* (BDSC_6409), *UAS-bskRNAi(II)* (BDSC_36643), *UAS-egrRNAi(II)*

(HMC) (BDSC_55276), *UAS-egrRNAi(II)(KK)* (VDRC_108814), *UAS-egrRNAi(III)* (VDRC_45253), *UAS-hid(II)* (BDSC_65403), *UAS-Myc(II)* (BDSC_9674), *UAS-RFP(II)* (BDSC_30556), *UAS-RFP(III)* (BDSC_31417), *ubi-GFP,FRT40A* (BDSC_5629).

*FRT40A,UAS-hid*, *FRT40A,hml-Gal4*, *FRT82B,ubi-GFP,hid^1*, *hml-Gal4,UAS-RFP*, *hml-Gal4,bsk^1*, *UAS-RFP,bsk^1*, *arm-Gal4,srpHemo-H2A.3XmCherry*, and *puc-lacZ,FRT82B,ubi-GFP* were generated by standard recombination methods. Recombinants were screened by visible phenotypes of *FRT40A, arm-Gal4 > UAS-hid, FRT82B, ubi-GFP, hid^1, hml-Gal4 > UAS-RFP, srpHemo-H2A.3XmCherry, puc-lacZ, bsk^1/bsk^1*.

Genotypes used in each figure are given in Table S1.

### Egg-laying assay
Prior to egg collection, we treated flies, which were previously kept in a 25 °C incubator, with $CO_2$ and then let them lay eggs for 1 h to get rid of old eggs. After that, eggs were collected for 1 h and then kept at 25 °C followed by other procedures below.

### Clonal analysis
*FRT82B,tub-HA:Myc* was used to generate *Myc* 'twin-spot' clones inducing cell competition as described by Steiger et al.[64]. Larvae carrying these transgenes were subjected to a 1-h heat shock at 37 °C to create twin-spot clones, and dissected at 16, 24, 48 and 72 h ACI.

*Minute* 'twin-spot' clones were generated as described by Li and Baker[65], but using FRT40A instead of FRT42. A single *Minute* mutation, M(2)Z, was used in this study. Larvae carrying transgenes were subjected to a 1-h heat shock at 37 °C to create twin-spot clones, and dissected at 72 h or 96 h ACI.

*dMyc* flip-out clones (with *Act > y>Gal4* driving UAS-*Myc*) were generated in a wild-type background, using a strategy analogous to that used by Moreno and Basler[51], to generate wild-type flip-out clones in a *dMyc* background. In Figs. 4, S8 and S15, larvae containing *Act > y>Gal4* transgene were subjected to heat shock for 10 min or 1 h at 37 °C to generate random GFP marked "flip-out" clones, and dissected 14 h ACI. For Figs. S14 and S17, larvae containing *Act > y>Gal4* transgene were subjected to heat shock for 7.5 min at 37 °C to generate random GFP marked "flip-out" clones, and dissected 48 h ACI.

Wild-type flip-out clones in a *dMyc* background (with *tub>Myc>Gal4* or *tub>Myc>LexA*) were generated as described by Moreno and Basler[51]. In Fig. S17D, H–K and M–O, larvae containing *tub>Myc>Gal4*, *tub > CD2>LexA* or *tub>Myc>LexA* transgene were subjected to heat shock for 10 min at 37 °C to generate random GFP marked "flip-out" clones. In Fig. S17C and E, larvae containing *tub>Myc>Gal4* transgene were subjected to heat shock for 7.5 min or 12.5 min, respectively, at 37 °C to generate random GFP marked "flip-out" clones. All these larvae were dissected 48 h ACI.

### Dissections and staining
Both female and male larvae were dissected in 3rd instar stage for imaging of imaginal discs except in the experiments shown in Fig. 6, for which only females were used (as alleles on both X chromosomes were required). Third instar wandering larvae were collected and dissected in ice-cold phosphate-buffered saline (PBS) and transferred directly into fix solution (4% paraformaldehyde and 0.05 M EGTA in PBS). Samples were fixed for 30 min at room temperature. Upon 5 washes (10 min each) in PBT (0.1% Tween-20 in PBS) on a rotor at room temperature, samples were stained with DAPI (4′,6-diamidino-2-phenylindole) before another 5 washes (10 min each) in PBT. For 'NoWash' discs, samples were fixed at room temperature for 30 min, and then fix solution was replaced by PBT. For discs that were collected for the wash test, samples were fixed for 30 min at room temperature and followed by 0, 1, 2, 3, 4, or 5 10-min washes with PBT, respectively. For discs that were washed thoroughly, samples were fixed for 20 min at room temperature, and then washed rigorously 7 times with PBT (10 min each) before stained with DAPI.

For immunostaining, same steps were repeated until 5 washes after fixation. Samples were blocked overnight at 4 °C in blocking solution (1% BSA, 0,3% Deoxycholate and 0.3% TritonX-100 in PBS). Afterwards, samples were incubated with primary antibodies diluted in blocking solution at 4 °C overnight, then washed 6 times in PBT (10 min each) and incubated with secondary antibodies and DAPI diluted in PBT for 1.5 h at room temperature on a rotor followed by another 5 washes in PBT (10 min each).

The following primary antibodies were used: mouse anti-β-galactosidase (Promega, Z3781) 1:1000, rabbit anti-Dcp1 (Cell Signaling Technology, 9578) 1:100, mouse anti-Patched (Developmental Studies Hybridoma Bank, *Drosophila* Ptc (Apa 1)) 1:1000, and mouse anti-Wingless (Developmental Studies Hybridoma Bank, 4D4) 1:1000. The following secondary antibodies were used: Alexa Fluor goat anti-mouse 488 (Thermo Fisher Scientific, A11001) 1:1000, Alexa Fluor goat anti-mouse 555 (Thermo Fisher Scientific, A21422) 1:1000, Alexa Fluor goat anti-rabbit 555 (Thermo Fisher Scientific, A21428) 1:1000, and Alexa Fluor goat anti-rabbit 647 (Thermo Fisher Scientific, A21244) 1:1000. 100 μg/mL DAPI (Invitrogen, D1306) were diluted 1:1000 in in PBT.

### Imaging
Imaginal discs were dissected from fixed samples and mounted in 50% glycerol on slides. Z-stack images of eye and wing discs were obtained with a Zeiss LSM 780 laser scanning confocal fluorescence microscope using 20x or 63x oil objective.

Images were analyzed using Fiji and processed using Adobe Illustrator.

### Quantifying cell competition
Area ratios were obtained by counting total numbers of above-threshold pixels using Fiji. To obtain values for bright clones, the threshold brightness of an image was adjusted manually to the point where only bright clones could be observed. Similarly, measurements for dark (e.g. GFP-negative) clones were obtained by performing the same steps after inverting the image. We verified that area ratio obtained by this method closely mimics that obtained by outlining clones manually and comparing clone areas (Fig. S2; clone area measurement was performed using Fiji Freehand Selections and Measure).

Quantifications of Dcp1-positive cells were made in maximal projections of wing imaginal discs using Fiji Cell Counter. Only Dcp1-positive cells within the wing pouch and hinge area were counted.

The sizes of flip-out clones were measured by using Fiji Trainable Weka Segmentation.

Statistical significance was calculated using the two-sided Mann-Whitney U test and Chi-squared test.

### Quantifying hemocyte recruitment and distribution
Hemocyte numbers were obtained from fluorescent images using Fiji Cell Counter. For Fig. 2E, slices containing hemocytes in wing discs carrying *dMyc* clones were imported into a Wolfram *Mathematica* notebook. GFP, RFP and DAPI channels for each slice were saved as TIFF files and imported with a coordinate for each pixel. After adjusting brightness, GFP and DAPI files were used to outline clonal boundaries, DAPI files were also used to outline disc edges, and RFP flies were utilized to determine the location of hemocytes. Distances between each hemocyte and the nearest clonal boundary were then calculated (10 discs were used for ACI72h and 8 were used for ACI48h). The data were converted to a spatial distribution histogram with 15 bins (bin size = 2 μm) and are plotted in pink in Fig. 2E. For comparison, we sought to generate an expected distribution under the null hypothesis, i.e. the hypothesis that hemocytes are distributed randomly. However, we noted the locations of hemocytes are generally not uniform in terms of distance to the disc edge, so to generate a more accurate null distribution we first divided up the same discs into eight concentric territories representing 20.76 μm bins away from the edge and measured the proportions of hemocytes that fell within each bin. Next, 1000 points were randomly seeded into the eight territories of one optical slice for each disc, with the number of points placed in each territory proportional to the number of hemocytes that was observed in each bin. This was repeated for all of the discs, with the total number of slices analyzed being 15 for ACI72h and 17 for ACI48h. Finally, the distance between each such "random" point and the nearest clonal boundary was calculated, just as had been done for the hemocytes in the same discs; the null distribution is plotted as blue bins in Fig. 2E. In addition to showing the expected and observed histograms, Fig. 2E also plots cumulative histogram values as solid lines. The hemocyte nucleus areas were quantified by using Fiji Trainable Weka Segmentation. Statistical significance was calculated using the two-sided Mann-Whitney U test.

### Plotting
In many figures, box and whisker plots were used to provide summary information about data sets. In such plots, the box represents the interquartile—spanning from the first quartile to the third quartile, with a white line inside marking the median—and the whiskers show the range of the data, reaching the minimum and maximum values.

### Reporting summary
Further information on research design is available in the Nature Portfolio Reporting Summary linked to this article.

## Data availability
All data generated or analyzed during this study are included in this published article (and its supplementary information files). Source data are provided with this paper.

## Code availability
Code used in data analysis for Fig. 2 is provided as Supplementary Code 1.

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

## Acknowledgements

We thank Nick Baker, Jose de Celis, Peter Gallant, Laura Johnston and Hiroshi Nakato for generously sharing strains and Daria Siekhaus and Nick Baker for helpful discussions. We also thank Martha Lopez and Bryan Alfonso Ramirez-Corona for their assistance. The authors wish to acknowledge the support of the Chao Family Comprehensive Cancer Center Optical Biology Shared Resource, supported by the National Cancer Institute of the National Institutes of Health under award number P30-CA062203. The work was funded by NIH grants P50-GM076516, U54-CA217378, and grants from the Cancer Research Coordinating Committee of the University of California and the Chao Family Comprehensive Cancer Center (A.D.L.) as well as NIH grant R01-HD095246 (Z.W.).

## Author contributions

A.L. supervised and Y.Z. designed and conducted experiments. Z.W. provided essential ideas and support for experimental design and execution. The manuscript was written by Y.Z. and A.L. and reviewed by Z.W.

## Competing interests

The authors declare no competing interests.
