## [Transparent Peer Review file · Nature Communications]

Epithelial cell competition is promoted by signaling from immune cells

Corresponding Author: Dr Arthur Lander

Version 0:

Reviewer comments:

Reviewer #1

(Remarks to the Author)

This manuscript identifies a contribution of the *Drosophila* innate immune cells— hemocytes- in cell competition models that manipulate *Myc* and *Minute* gene dosage in the developing eye and wing primordium of the fly. Strikingly, the authors show that the presence of hemocytes is necessary for the removal of loser cells: an ablation of the hemocyte population leads to a full rescue in terms of clone size for *Myc* cell competition. While hemocyte recruitment appears to be independent of JNK stress signaling or cell death in tissues that undergo cell competition, it is instead dependent on the juxtaposition of different genotypes present within one tissue. The authors identify that hemocytes contribute the TNF ligand *Eiger*, which is necessary for *Myc* cell competition. Then the authors show that cell death is partially reduced in *Myc* cell competition when the hemocyte population is ablated.

The authors present exciting data that hemocytes are required for cell competition, which contrasts with previous literature on this closely studied and important process. There is also an interesting perspective, concerning the relation of the physiological function of innate immune cells in non-cancerous bodies to those in cancer. The discussion point about hemocyte involvement in a bistable switch is useful. This message is potentially appropriate for Nature Communications if well-supported. In its current form, the manuscript does not present a clear working model that is fully supported by experimental data.

Below you will find the following main concerns:

1) The authors state that 'These results suggest that tumor surveillance and cell competition are, in fact, a single phenomenon' (line 78-79). This is not supported by the data. In fact, types of cell competition have various, stark differences to tumor surveillance. The authors do not present data that is inconsistent with their model, and do not acknowledge published data that predates that model. The authors cite a study from the Vidal lab concerning tumor surveillance, but do not mention major differences in terms of hemocyte involvement: 1) direct attachment of hemocytes to dying cells, 2) juxtacrine *Eiger* release rather than the proposed disc-wide release, and critically 3) the absence of juxtaposed WT and mutant cells. This causes a small decrease in tumor size but at a late stage of tumorigenesis that cannot really be considered tumor surveillance (Reference 33). The authors do not cite a more appropriate study last year from the Bilder lab which fits better with tumor surveillance and cell competition: it studied small oncogenic clones juxtaposed to WT cells that are eliminated before they can form a tumor mass. This is mediated by circulating *Eiger* from the fat body but does not involve hemocyte *Eiger*. Overall the manuscript would benefit from clear distinctions between different types of cell competition.

2) The authors use a *Myc* cell competition model for most experiments but only check a few with a *Minute* cell competition model as well (Figure 1, Figure S6). I believe that key experiments should be addressed with *Minute* cells to justify a generalized claim, as the title of the manuscript indicates. Please test whether hemocyte-produced *Eiger* is important for *Minute* cell competition and that cell death in *Minute* clones is reduced once hemocytes are ablated. The latter is particularly important because the Moreno lab previously showed that hemocyte ablation by *He>hid* increased, not decreased, the number of C3 positive cells in *Minute* disc (Reference 30), which would be the opposite of the author's model. Alternatively, the authors could tone-down the generality and use the term *Myc* cell competition more specifically.

3) The authors use a combination of *HmlΔ-Gal4* and *UAS-hid* to ablate hemocytes. However, a study from the Theopold lab

shows that expression of apoptotic inducers by HmlΔ-Gal4 may increase overall hemocyte numbers that are not hml-positive (Arefin et al., PLoS one, 2015, doi:10.1371/journal.pone.0136593). The authors should show the efficiency of hemocyte ablation using established hemocyte markers such as *srp*, *Hemese*, or *NimC1*. Furthermore, could hemocyte ablation have secondary effects, e.g. on antimicrobial peptide production as seen in the Theopold lab paper, which is known to regulate cell competition?

4) The authors conclude that mosaic clones recruit hemocytes to the wing primordium. However, the data are not fully convincing that cell competition sites recruit hemocytes and that disc-attached hemocytes are functionally important. This is based on these three observations from the manuscript:

1) In 'No Wash' condition, the number of disc-attached hemocytes seem to be unchanged between control and Myc conditions (Fig. S7Q), which does not seem consistent with the author's argument that hemocytes are associated specifically with discs undergoing cell competition.

2) Hemocytes appear to accumulate on a defined locus (anterior-, ventral-facing portion of the disc), which is inconsistent with the idea that hemocytes are recruited to sites of cell competition.

3) Hemocyte attachment on Myc mosaic discs is very variable and some Myc discs have no hemocytes attached (Fig. S7Q). However, hemocyte ablation very consistently results in a lack of competition. Together this implies that the attachment of hemocytes onto discs may not be necessary for hemocytes to act on discs that undergo Myc cell competition.

These data seem to challenge the analogy with the tumor surveillance shown by the Vidal lab.

5) The mechanism that is presented appears incomplete and does not lead to a clear model on how hemocytes affect Myc cell competition. Three main concerns could be addressed experimentally to clarify the author's model:

1) What is the role of hemocyte-originated Eiger and where does the TNF ligand bind in Myc cell competition? *Drosophila* has two TNFR receptors, Wengen and Grindelwald. At least for Grindelwald, the Bilder lab showed that the receptor is exclusively located on the apical side of cells in the epithelial wing disc tissue (de Vreede et al., Science, 2022, doi: 10.1126/science.abl4213). Therefore, it seems that the receptor would not be able to directly bind the TNF ligand from hemocytes in proximity, unless the disc was disrupted somehow..

2) The authors show a complete clone size rescue for Myc cell competition when hemocytes are ablated (Fig. 1). However, the effect on cell death in this condition is minor (Fig. 7). How do the authors reconcile this? They could analyze other factors that may be changed by an ablation of hemocytes in Myc cell competition, such as growth rate or JNK activation changed in loser cells.

Below you will find minor concerns:

1) In the title, it is unclear what 'directed' is supposed to mean. The model I think proposes that hemocytes are more permissive than instructive for competition. Please revise for a more appropriate description.

2) Add clear labeling to all figures, e.g. Fig. 7 misses Cas3, GFP, DAPI labeling.

3) Is hemocytes adherence to eye imaginal discs similar to wings, in that the difference is seen only with an intermediate number of washes? Please clarify.

4) Please clarify the statistical analysis used. What do the solid bars and whiskers on the graphs mean? Add a supplemental table with all p-values and statistical tests listed as well as a proper labeling of graph error bars in the figure legends. Use sufficient numbers of samples for experiments— some experiments, as for example Fig. 7A and Fig. 7C, use only n=5. This is unfortunately not sufficient to make a solid conclusion.

5) Does the number of Caspase3-positive cells change in Myc; Hml>egr RNAi and Myc; Hml>bskDN?

6) line 42: The sentence is incorrect. The Pastor-Pareja lab used *dlg* mutant discs that do not show cells in juxtaposition (Ref. 27).

7) line 89-107: Please cite any previous papers that developed these systems and showed that they induce cell competition; if this approach is original, perhaps state why the system used here is different than previous work. It would help a lot to add a Supplementary Figure that diagrams the clonal induction: which clones are generated in Myc and Minute cell competition models and how are they marked (GFP+, GFP-, lacZ+, lacZ-).

8) line 138: Why is it notable that hemocytes recruitment is 'early'? This should be a time when cells are being eliminated and expressing Caspase 3.

9) Fig. 4 shows the same experiment with the reporter either inserted on 2nd or 3rd chromosome. Please move one of the experiments to the Supplementary Figures.

- 10) line 223: Please explain the discrepancy with the study from the Johnston lab (ref. 59).
- 11) Fig. 7: Is apoptosis restricted to the low Myc-expressing cells? Is apoptosis localized at the clone boundary mainly?
- 12) Fig. S3E: Why is the shown disc either earlier in development or at a lower magnification?
- 13) Fig. 1F: Why are some lacZ⁺ cells less bright and hazy than others in Minute; hemocyte-ablated disc? It seems that there are two classes of cells.
- 14) Please specifically address why the authors' results conflict with those of the Moreno lab (ref. 30) for hemocyte requirement on death of cells.

Enhancements for manuscript:

How are hemocytes attracted to the tissue undergoing cell competition, if at all? The authors show in Fig. 4 that hemocytes are attracted through cell juxtapositions. In the discussion (line 314-316), the authors mention that ROS or basement membrane degradation may lead to the attraction. They could address whether a correlation between ROS or basement membrane degradation and hemocyte attachment is present in Myc cell competition.

Reviewer #2

(Remarks to the Author)

This work by Zhu et al challenges the traditional perspective by proposing that tumor surveillance is not primarily designed for attacking cancers but rather represents a broader strategy to ensure conformity of epithelial cell behaviors.

Tumor Surveillance refers to the body's natural defense mechanism to identify and eliminate abnormal or cancerous cells that may form tumors. The authors found that during cell competition, hemocytes are recruited independent of cell killing and JNK activation. They support that both cell killing and cell competition depend upon Eiger (TNF ligand) production specifically by hemocytes.

This work is interesting. Below are my comments that I consider necessary to be resolved to strengthen the authors' conclusions. Please, as I mention also in the end, state clearly which conclusions have been shown for the Minute competition.

Major concerns.

- 1) In the experiments depicted in Figure 1, how long was the egg laying of the crosses? Can the authors provide more information regarding the protocol for the generation of clones at different time points? (Egg laying, heat shock time, and heat shock duration). Do the authors find any developmental delay when they ablated hemocytes? For example, in Figure 1E the disc looks younger than in 1F. This is more important for Minute competition, where the twin spot (Minute/Minute) dies therefore there is no clear indication of the recombination event. For example, the difference in the area of the "dark clone" in Figure S2 could be due to the different developmental stages that the heat shock was applied.
- 2) Have the authors stained for apoptotic markers in genotypes of Figure 1 to check if apoptosis of Minute loser cells is reduced upon hemocyte depletion? For Myc dependent result upon hemocyte depletion shown in Figure 7, my comments are below.
- 3) In Figure S3 the discs have different sizes. Is that due to differences in imaging or this implies differences due to developmental stages? If this is the case, the question that comes to my mind if there is any difference in hemocyte attachment at different stages since the folding of the wing discs is increasing in the late 3rd instar larva and the authors state that they have observed accumulation of hemocytes associated with folds (line 133)
- 4) In Figure S7, where the authors support that washes affect the hemocyte attachment, I am concerned since the difference is obvious in 4 washes, while it is not obvious (not statistically significant) in 2, 3, or 5 washes. If each dot depicts one disc, in 5 washes for example 4 discs of control behave as 3 discs of myc discs and there are only 2 discs that show an increased number of attached hemocytes. Here, my concern is also related to the developmental stages of the discs and how this could affect the attachment of the hemocytes. If each dot is one disc, I recommend that the number of discs in each wash should be increased. Especially since this result is contradicting an earlier study (#58 Reference) and also supports that hemocyte attachment is affected during competition.
- 5) In Figure S8B the ratio dark/bright is higher in the eye imaginal disc, which does not occur in S8C when p35 is expressed by GMR driver. Have the authors checked whether in eye discs their Myc super competition system supports the death of losers when are in close proximity to 2xmyc clones? GMR is expressed posterior to morphogenetic furrow (MF) (therefore mainly in post-mitotic cells), but not in the anterior to MF proliferative cells. How the competition anterior to MF is rescued by GMR-p35 since the competition in proliferative cells will still impact the final sizes of the clones? Have the author checked the hemocyte recruitment and the clonal areas in the eye discs in genotypes shown in Figure 3D-F, where apoptosis is blocked by hid1 heterozygosity?
- 6) In Figure S9 the authors argue that cell competition happens when dMyc flip-out clones are generated in a wild-type background. Please provide an apoptotic marker for these genotypes, to show that cell death is induced outside the myc overexpressing clones. This is necessary since their conclusion for hemocyte recruitment due to clonal juxtapositions depends on proving that this system behaves as a supercompetitor system.
- 7) Have the authors checked the hemocyte recruitment when they have two copies of tub-HA:Mycw⁺ transgene (genotype: FRT82B, tub-HA:Mycw⁺/ FRT82B, tub-HA:Mycw⁺). I believe this will strengthen their conclusion that "it is not dMyc overexpression per se, but rather the juxtaposition of cells with different levels of dMyc, that drives hemocyte recruitment".
- 8) The authors in Figure 5 present a requirement of egr by hemocytes for myc-dependent cell competition, by measuring the growth of clones during myc super competition. Could the authors provide the caspase-3 staining in those genotypes

presented in Figure 5.

9) One also important conclusion in the paper is shown in Figure 6, where *hml>bskRNAi* and *hml>bskN* is sufficient to block myc-dependent competition (estimated by area ratio of the clones). Please provide representative images of these experiments.

10) The effect of JNK activation in hemocytes during myc-dependent competition will be strengthened by checking apoptosis in the genotypes of Figure 6. Could the authors rescue the myc-dependent apoptosis when they deplete *bsk* or overexpress *bskDN* in hemocytes (by *hml* driver)? Similar to what they support in Figure 7, where they kill the hemocytes.

11) In Figure 7 panel F, the black clones have 2 extra copies of *Myc* and act as supercompetitors. Please provide a "zoom-in" of panel F, since the disc that is shown gives the impression that a significant part of apoptosis is localized in the black areas. Also, the cell death localization of double GFP clones (losers, with no extra *Myc*) in the anterior compartment looks quite away of the boundaries.

12) In Figure 7B the cell death is quite a lot. Have the authors checked the cell death in the genotype of 7B without heat shock (*hsFLP*; *FRT82B*, *ubiGFP/FRT82B*, *tub-HA:MycW+*)?

Minor concerns

1) In Figure 1D, the twin spot clones have two copies of *arm-lacZ* and the background has only one. It is not obvious in this image the difference in the background with the twin spots (1 vs 2 copies *arm-lacZ*). How long was the heat shock applied in such experiments?

2) Please state clearly the conclusions that cover both the cell competition system (*myc* or *Minute*) and the conclusions for only the *Myc* system. For example, the *Minute* competition was used only for two conclusions: a) ablation of hemocytes leads to higher occupation of the winner area compared to *Minutes*, and b) hemocytes are recruited to sites of competition. All the other results have not been shown for the *Minute* competition. This should be clear in the title of the paper (or in the abstract), in the title of the result section, and in the discussion. E.g. Hemocyte recruitment occurs regardless of whether cells die or competition succeeds in myc super competition, Expression of *Eiger/TNF* by hemocytes is necessary for myc-dependent cell competition, Recruitment of hemocytes during myc competition does not require JNK signaling, Hemocytes contribute to killing loser cells in *Myc* super competition. I was kind of puzzled about which conclusions have been shown in *Minutes*.

3) Please also provide references where the same fly stocks (*FRT82B*, *tub-HA:Myc*) have been used in super-competition

Thank you

Reviewer #3

(Remarks to the Author)

The manuscript by Zhu et al. describes the roles of innate immune cells, called hemocytes, in modulating the process of cell completion in the *Drosophila* imaginal discs, by analogy to tumor surveillance with immune cells. The authors show that hemocytes are recruited to the places, where cell completion occurs, in an apoptosis-independent manner, but their recruitment is limited in more peripheral regions of the wing disc than centrally. The authors also show that *Eiger/TNF* that is produced by hemocytes contributes to cell completion at least in part.

This is an interesting study, suggesting that tumor surveillance and cell competition are mechanistically related processes. However, it seems that hemocytes just help trigger apoptosis in loser cells to occur only in the peripheral part of the wing disc, whereby hemocytes preferably locate. Although the hemocytes produce *Eiger/TNF*, it is the same signal as during epithelial cell competition, it is unclear to what extent hemocytes-mediated cell killing contributes to cell competition. Thus, this study does not provide a novel mechanistic sight into cell competition or tumor surveillance.

The major points are as below.

1. Figure S5 shows hemocytes are attracted via cell competition only in the peripheral part of the imaginal disc. How do the authors explain this? What is the signal from the winners/losers attracting hemocytes?
2. In Figure 5, is it significantly different between the groups: *Myc*;hemocyte-ablated vs *Myc*;egrRNAi? If yes, that suggests another factor potentially mediating this process.
3. In Figure 7, is it significantly different between the groups: control vs *Myc*;hemocyte-ablated? A simple interpretation for this is that the contribution of hemocytes to *Myc*-induced cell killing is only in part.
4. In summary, the conclusive remark "...tumor surveillance is less a mechanism evolved to attack cancers than a manifestation of a general strategy for ensuring conformity of epithelial cell behaviors." is difficult to understand in relation to this study.

Version 1:

Reviewer comments:

Reviewer #1

(Remarks to the Author)

The authors have done extensive experiments and text revisions to address the concerns raised by the reviewers. Their efforts reinforce two valuable contributions. A- that hemocytes play a role in modes of cell competition where they were thought not to. B- that Eiger from hemocytes is important. These findings should be published.

However, this revised manuscript highlights remaining concerns with the experiments that are not acknowledged. The current presentation would add confusion to the field especially for readers of Nature Communications who are not specialists in cell competition. I recommend that the text be again revised to provide a more transparent and balanced account. For this reason, this re-review remains lengthy.

Major comment 1: 'tumor-suppressive cell competition (TSCC)' instead of 'tumor surveillance'

The manuscript states that 'Drosophila displays robust tumor surveillance, which we define as recognition and elimination of tumor cells shortly after they first arise. This process is illustrated by the fact that small mosaic clones that are mutant for tumor suppressor genes, such as scribble, dlg1 and l(2)gl, when created in the imaginal discs of larvae, are efficiently killed and actively extruded, whereas large, contiguous groups of the same mutant cells grow uncontrollably¹⁹⁻²⁶'

The authors use of the term 'tumor surveillance' to describe the killing of tumor suppressor gene mutant cells from imaginal discs creates a lot of confusion and misleading to general readers of the article, who may falsely equate the term 'tumor surveillance' with 'tumor immune surveillance'. The term 'tumor surveillance' is already widely used in the mammalian tumor immunology field and involves an active role of circulating immune cells in recognizing and killing tumor cells. For removal of scribble defective cells by non-immune means in mammalian tissue culture, the Fujita lab has used the term Epithelial Defense Against Cancer. In flies, the killing of tumor suppressor gene mutant cells from imaginal discs has usually been called cell competition. Although the Igaki lab once used the term Intrinsic Tumor Suppression, they more recently have carefully chosen 'tumor-suppressive cell competition (TSCC)'. While it is true that the Vidal lab shows that hemocytes are involved in this, the Bilder lab showed evidence for involvement of cells that are neither immune nor epithelial. The authors themselves are not consistent with its use. At 218, the authors call killing of tumor suppressor gene mutant cells cell competition and lump it together with Myc overexpression competition.

Overall, the authors introduce a new term which further muddies the distinction between the various situations and prevents comparison to the established body of literature. The authors use rhetoric to say in the introduction that fly 'tumor surveillance' and cell competition are thought to be distinct, but then conclude that actually they are not that distinct. Both of these views are overstated 'straw-men'. The authors should say clearly and precisely what the scientific advance is and not equate their findings with 'tumor immune surveillance', which is a different concept entirely. They should revise the manuscript and abstract accordingly.

Major comment 2: The Myc model used is atypical and its interpretation is complex.

Two reviewers requested that the authors provide explicit illustration and citation of their main technique to make Myc overexpressing cells. Figure S1 now makes it clear that they do not use a well-defined system for cell competition. This approach by the Gallant lab, used perhaps only once, creates three genotypes of cells, rather than two genotypes which all standard cell competition systems generate. It is for this reason that the Johnston lab in 2014 developed the system that is since almost universally used for Myc competition and is now shown for one assay for Fig. S13. In the Gallant lab system three genotypes of cells are created and also three types of borders are created that will differ with each clone. But the competition quantitation measures only the ratio of 2X extra Myc to no extra Myc. Figure S1B is a bit misleading because the Minute homozygous cells do not grow and are killed immediately.

While this does not make the findings of the paper invalid, it is a limitation in linking the findings to the rest of the literature that uses the Johnston Myc competition system. For instance is the increase in ratio from excess growth of the 2X myc cells relative to the 1X myc cells, or from apoptosis of the 0x Myc? In Figure 7G: there are many clones that show no CC3 staining, and also the CC3 staining does not show any pattern with respect to clone genotype or the three different types of clonal boundaries. The authors response to Reviewer 2 Major Concern 11 is insufficient and needs to be quantitated for the system that they use.

I request that the authors use the Johnston lab system to test the two main advances claimed: - that hemocytes play a role and that Eiger from hemocytes is important.

Major comment 3: No evidence of the functional importance of disc-attached hemocytes

The authors raised concerns about a previous comment (R1 major concern 4), which questioned the functional importance of hemocyte attachment. We appreciate that the authors conducted additional eye-disc washing experiment (Fig S10, S11), which confirm their wing disc observations. However, the authors have not shown the functional importance of disc-attached hemocytes, which undermines the importance of Fig. 2, 3 and 4. Below are our specific comments.

1) (Previous comment) In 'No Wash' condition, the number of disc-attached hemocytes seem to be unchanged between control and Myc conditions.

The authors argue that this is due to non-specifically bound hemocytes. Assuming this is the case, the authors cannot argue that hemocytes are 'recruited'. The authors should address this through text changes by toning down the role of 'recruitment' and replace with 'attachment' or 'increased adherence'. Instead, one may interpret these hemocytes as 'stickier' (less detached from disc upon washing). Given the model where hemocytes elevate Egr levels to promote cell competition, it is unclear whether it is only the 'stickier' hemocytes that are the important cells here.

2) (Previous comment) Hemocyte attachment on Myc mosaic discs is very variable and some Myc discs have no hemocytes attached.

The authors replied in their rebuttal that '-what we routinely observe after 4-5 washes probably reflects the removal of some specifically bound hemocytes, so we cannot be confident that a disc in which we didn't observe hemocytes lacked any specifically bound ones at the time of dissection'

While it makes sense that additional washes may remove hemocytes, many Myc wing discs have zero or almost zero hemocytes even in 'No Wash' conditions (Fig S10Q: 3/6=50%). Yet, hemocyte ablation consistently (100%) results in a lack of competition (Fig 1G). This observation casts doubt on the interpretation that attachment on discs is indispensable for

hemocytes to regulate Myc cell competition.

The authors further replied that 'hemocytes are highly motile cells and likely move on and off discs throughout the experimental period.'

This is an interesting perspective, and it is possible. However, the authors show in detail that a population of hemocytes becomes 'stickier' and attach to the discs even after several washes. It would be expected to see that these 'stickier' hemocytes attach to the disc long-term, so that a change would be quantifiable.

The authors should at least acknowledge the limitations of their interpretation.

Major comment 4: Difficulties in interpreting Caspase-3 as an apoptosis readout

The authors point out that Caspase-3 staining has limitations as a direct readout of apoptosis. The author's issues are summarized here: 1) hemocyte ablation impacts the clearance of cell debris; 2) the length of clonal boundaries impacts the amount of apoptosis per disc in cell competition context; 3) not all Caspase-3-positive cells undergo cell death; and 4) Myc-overexpression leads to cell-autonomous apoptosis. However, apoptosis measurements are –next to clonal area measurements– one of the essential readouts for cell competition.

Other studies have mastered the listed hardships to produce reliable apoptosis measurements. Each of these issues can be addressed: 1) Apoptotic cells are extruded basally (hemocyte-independent) and the debris is then cleared through phagocytosis by hemocytes. Lolo et al. have faced this issue during their study and observed that the un-cleared debris accumulates specifically on the basal side of the disc, below the epithelial layer. The authors could address this issue by comparing different sections of the disc, e.g. further apical as well as further basal. 2) The authors can address a correlation between clonal boundary length and apoptosis by measuring the length of clonal boundaries and measure adjacent apoptosis, in a similar fashion to the measurements done by Yamamoto et al. previously ('dying cells per clone perimeter', doi.org/10.1038/nature21033). 3) The authors could test alternative readouts of apoptosis, such as TUNEL assay, to avoid non-apoptotic roles of Caspase-3. 4) While cell-autonomous apoptosis is a contributor to the dying cells in discs, this should remain constant across the tested Myc-overexpression contexts in this study.

Major comment 5: Imbalanced presentation of preferred model

The text also overweighs the authors' preferred model and does not acknowledge problems in interpretation. As long as the data do not exclude the authors' model, as long as there are possibilities that could explain the differences of this model with published papers, their model is a winner. One example of this is line 279. The results with Minute cells are not statistically significant. The authors provide two explanations why this might be artifactual, but do not state simply that Minute cells may work by a different mechanism. In Figure 3 and response to Reviewer 2. GMR>P35 should not effectively block cell competition in this system –most competition should take place in the proliferating cells ahead of the furrow and not in the SMW behind it. So this genotype does not test whether hemocyte recruitment depends on apoptosis –there will still be extensive competitive apoptosis in the discs. Additionally, any cell blocked by p35 will be 'undead' and recruit hemocytes by a mechanism shown by the Bergmann lab, confusing interpretation.

Minor comments:

41: the role of systemic factors including Egr and insulin should be included

116: the authors explain discrepancy with the Moreno lab partly by claiming that their hemocyte elimination technique is more efficient. They show that it eliminates hemocytes associated with discs, but does it eliminate them from the entire larva? The model proposed apparently does not require hemocytes to be near to the clone.

347-361: how does this discussion relate to the authors' data that recruitment of hemocytes is not dependent on JNK signaling?

The pJNK AB in Fig S15 is not convincing. disc A seems to be imaged basally and there is more JNK activation than expected. In B the staining does not seem to correlate with clone genotypes or clone boundaries and there is clear staining in winner as well as loser clones.

299, what do the authors mean by the distinction between cell killing and cell competition?

Reviewer #2

(Remarks to the Author)

I am generally satisfied by the replies of the authors in my previous comments.

I have one major concern regarding the Minute competition.

In Figure S20, where they have generated wild-type cells in the Minute background, the caspase 3 staining stains the wild-type cells and not the Minute cells. Also in the same figure when they ablate hemocytes the caspase staining is increased in the wild-type area. These results weakens the conclusions for the Minute competition

I am positive about the author's conclusions regarding the Myc super competition, but not the Minute competition.

Best wishes,

Reviewer #3

(Remarks to the Author)

The revised manuscript by Zhu et al. has strengthened their data and arguments by adding more detailed examination of minute cell competition and integrating the published papers in relation to cell competition and tumor surveillance.

The conclusive remark "These findings support a view of tumor surveillance as less a specific mechanism for eliminating

cancer than a general process for promoting phenotypic uniformity among epithelial cells.” is now supported by the data. However, the authors have not fully explored how to interpret the indication from the data in Figure 1 “both dMyc-associated and Minute-associated cell competition requires the presence of hemocytes”, based on the experimental evidence. Mechanistically, although hemocyte-derived Eiger/TNF promotes epithelial cell competition, there is more likely another key signal from hemocytes to explain the strong effect on cell killing in Figure 1.

This may be beyond the scope of this manuscript but the absence of a novel mechanistic view might make this work less attractive to broader readership.

Version 2:

Reviewer comments:

Reviewer #1

(Remarks to the Author)

I take no pleasure in continuing to argue with the authors, and also regret the time involved on both our parts. The work is a valuable contribution. But I can't say that the way this paper is written is OK. The terminology using 'intrinsic tumor suppression' is wrong, and this still leads to a straw man model that overstates the distinction between two types of what most people describe as cell competition.

- The authors have shifted to calling elimination of cells mutant for fly tumor suppressors such as scribble 'intrinsic tumor suppression'. The authors first use this term on line 49 without defining it. Intrinsic tumor suppression as far as I can see was defined by Scott Lowe in a 2004 Nature piece to refer to cell-intrinsic factors (e.g. oncogenes that also trigger autonomous cell death). The Igaki lab in 2009 used the term to explain what most others described as cell competition, but subsequently dropped it. The 2010 Vidal lab paper showed that this elimination is not intrinsic but involves hemocytes. So it is contradictory to continue to call this intrinsic tumor suppression, as the authors do again in the discussion:

Immune cells are thought to
327 play an active role in intrinsic tumor suppression²⁵

- It's demonstrable from the literature that investigators have overwhelmingly referred to this type elimination as 'cell competition', since it involves the killing of cells in mosaic genotypes but not uniform non-juxtaposed genotypes. That is the definition of cell competition, and the long list of common features has led many/most researchers to group the two into one phenomenon.

Similarities between intrinsic tumor suppression and cell competition include a requirement for
59 the juxtaposition of phenotypically different cells; involvement of Eiger, JNK signaling, and
60 suppression of Hippo signaling; and a process of cell elimination that involves cell killing and
61 active extrusion from the epithelium; these similarities have led many authors to refer to intrinsic
62 tumor suppression as a “form” of cell competition^{27,35,56}

- Clear evidence for the problems of this approach can be seen in that the authors cite the same data in the same paragraph to refer to scrib cell elimination first as 'cell competition' and immediately after as 'intrinsic tumor suppression'.

The secreted TNF ortholog Eiger potently activates JNK signaling in *Drosophila* (just as TNF
224 is a potent JNK agonist in mammals), and in flies mutant for *eiger*, cell competition is abrogated²⁴.
225 The cellular source of Eiger during cell competition is currently unclear. Disc epithelial cells
226 express Eiger at very low levels, and a significant increase in expression is not seen in discs
227 undergoing Myc-induced cell competition⁵⁸. Nevertheless, disc-derived Eiger has been implicated
228 in intrinsic tumor suppression²⁴, as well as Eiger derived from other sources. For example, *eiger*
229 is highly expressed by stimulated hemocytes^{25,74} (just as TNF is a major secreted product of
230 activated macrophages) and hemocyte *eiger* has been implicated in intrinsic tumor suppression
231 ²⁵, although recent evidence suggests that the fat body, which releases Eiger into the circulation,
232 may be the primary source of Eiger during tumor suppression³².

- I think on balance the term 'tumor suppressive cell competition', or perhaps 'structural cell competition' from Claveria and Torres Ann Rev 2016 is more accurate, unless the authors want to credit the argument from the Bilder lab that scrib cell elimination does not rely on juxtaposed genotypes at all. In the rebuttal, the authors say they don't want to use such terms because it implies that it and cell competition are already known to be the same thing. But that is in fact what most people in the field believe: that although there are interesting distinctions, the two are quite close-- as supported by the fact that most reviews (too many to list, but the authors must recognize this) group them together.

- To reiterate, this exaggerated distinction sets up a straw man that leads them to fundamentally misleading sentences like the following important paragraph which concludes their introduction. This weak approach is confusing and does disservice to the authors' valuable data.

These results suggest that intrinsic tumor suppression and cell competition are more closely
80 related phenomena than previously thought. This, in turn, suggests that the former may not be a
81 dedicated mechanism for attacking cancers, but rather a collaborative, physiological response of
82 epithelial and innate immune cells for dealing with phenotypic mismatches in general—a response
83 that sometimes happens to eliminate tumor cells. To this end, efforts to improve tumor
84 surveillance in human populations might be well served by a deeper understanding of cell

85 competition—in particular, how winner and loser fates get assigned, and how innate immune cells
86 are attracted to sites where cell phenotypes clash

• One more example of an inaccurate presentation that creates a false distinction: it has been shown that scrib cells are also intrinsically slower growing, just like Minute/+ cells.

If cell competition and tumor suppression share so many features, one might question why, in
336 the former case, losers tend to be slower growing than winners, whereas in the latter case cells
337 with the propensity to overgrow get eliminated.

I've made my arguments and will leave it to the Editor and Authors about how to proceed.

Other notes:

• I and the field will appreciate the authors' extensive additional work to clarify cell death with the better DCP1 assay.

• I and the field will also appreciate the authors repeating some experiments with the frequently used WT FLPout in Myc+ background model (S17). They find the elimination of hemocytes to play a significant role here, but it must be said that the effect size is pretty small and much smaller than the effects seen in Fig. 1. I would suggest that the authors make the 'median' bars visible in figures like this, they are almost impossible to see, so effect sizes are masked.

• Fig. S1: I suggest adopting a terminology such as Myc+ to distinguish the extra copies of Myc from the loss of a copy of Minute. The existing figure will suggest to most that a Myc/GFP cell is heterozygous loss of function. Also, thank you for adding in the figure that Minute homozygous cells are killed.

• is the below accurate? It looks to me that the difference between WT and Hml>egrRNAi in L is smaller than the difference between WT and clone>egrRNAi in P.

Here we obtained evidence that, in cell competition, Eiger

352 derived from hemocytes plays a major role, but Eiger from loser cells can also contribute to a
353 minor degree (Fig. S17).

Reviewer #2

(Remarks to the Author)

I would like to thank the authors for the experiments provided on this revised version.

I have few comments on Figure S22 for the experiment performed for Minute competition.

a) In Figure S22A while the clones (FRT40/FRT40) are obvious, the difference of twin spot (armlacZ, FRT40/ armlacZ, FRT40) versus the unrecombined background (alz FRT40/FRT40) is not obvious.

b) The effect of hml>hid (Figure S 22C) and of hml>egrRNAi (Figure S22D) on Dcp1 positive cells is really impressive (Figure S22F), while the effect on area ratio is not so. Why is that?

c) In the same experiment do the authors know if the UAS-hid and hmlGal4 transgenes are on the left or the right arm of the 2nd chromosome. If they are on the left arm, please provide the control experiments (hsFLP; UAS-hid FRT40/ arm-lacZ FRT40) and (hsFLP; hmlGal4 FRT40/ arm-lacZ FRT40) to make sure that the wild type cells (UAS-hid FRT40/UAS-hid FRT40) and (hmlGal4 FRT40/ hmlGal4FRT40) generated in Figure 22C and Figure 22D do not present any growth disadvantage that could interfere with the Minute competition.

In the material and methods the authors state that "Minute 'twin-spot' clones were generated as described by Li and Baker65, but using FRT40A instead of FRT42. Larvae carrying transgenes were subjected to a 1-h heat shock at 37°C to create twin-spot clones, and dissected at 96 h ACl." Why the authors did not perform dissection at 60-72 hrs after heat shock as it is usually performed for the Minute competition. When did they perform the heat shock (how many hours after egg laying)? The authors have provided an experiment showing that hemocyte ablation by hmlGal4>hid does not delay normal development (Figure S4). Have they noticed any delay upon hemocyte depletion when the flies are Minute? Are the dissections in Figure S22A-D performed on the same time after egg laying and heat shock? Can the authors provide an experiment with hml>hid depletion using the standard conditions that are used for the Minute competition, eg dissection after 60-72 hrs?

In Figure S5A while the clones (FRT40/FRT40) are obvious, the difference of twin spot (armlacZ, FRT40/ armlacZ, FRT40) versus the unrecombined background (alz FRT40/FRT40) is not obvious.

MINOR Comments

Since they are different myc competition assays, it would be helpful if in each image was more evident which system was used. For example for Figure S17

B. UAS-myc clones/WT background

C-G. WT clones/ tyb>myc background

I. WT in tub>myc background

J. WT in tub>myc background, hml>hid

K. WT in tub>myc background, hml>egrRNAi

M. WT in tub>myc background

N. egrRNAi in tub>myc background,

O. egrRNAi in tub>myc background, hml>egrRNAi

Thank you

Reviewer #3

(Remarks to the Author)

The second revision of the manuscript by Zhu et al. has addressed the concerns raised by the reviewers to a satisfactory degree.

However, it seems that the title "Epithelial cell competition is promoted by signaling from immune cells" does not reflect the description of the key findings and conclusive remark in summary, as below.

"Here, we challenge this view, showing that, in the fruit fly *Drosophila*, innate immune cells play much the same roles in cell competition as they have been shown to do in intrinsic tumor suppression. These findings support a view of tumor suppression as less a specific mechanism for eliminating cancer than a general process for promoting phenotypic uniformity among epithelial cells."

Thus, at least to this reviewer, the key findings and conclusive remark would be insufficient for attracting a broad readership. Thus, I hesitate to support this manuscript for publication in *Nature Communications*.

Version 3:

Reviewer comments:

Reviewer #1

(Remarks to the Author)

Thanks to the authors for engaging thoughtfully with the critique, which I think has clarified the paper's contribution.

One detail I noticed in Fig. 3c: I was not aware that heterozygosity for *hid* alone, rather than the H99 deficiency that removes multiple apoptotic regulators, was sufficient to suppress cell death. The authors may want to add a reference that supports that, and also make clear at the top panel that this is *hid* [1]⁺.

Reviewer #2

(Remarks to the Author)

I would like to thank the authors for their responses.

Major concern:

1) In control experiments of panel S22A, S22B, S22C, S22G, S22H and S22I the twin spots are not obvious, while they should be present and have similar size with clones.

2) In the experiment provided in S22K and S22L images, the staining for the clones (*b*-galactosidase) presents 3 different cell types (here we expect only 2). One does not have *lacZ* (wild type cells), one has lower *lacZ* staining (*Rp*⁺ cells) and the third cell type seems brighter (like having two copies of *lacZ*). We do not expect cells with 2 copies of *lacZ* to be viable, since they lack both *Rp* copies.

Authors should provide a better disc for these panels.

I would prefer the authors to tone down their conclusions for Minute competition or at least mention that only one Minute mutation was studied.

Best wishes.

Reviewer #3

(Remarks to the Author)

The third revision of the manuscript by Zhu et al. made the summary clearer. As the authors claimed, this could attract a broader audience.

Epithelial cell competition is promoted by signaling from immune cells

Zhu, Wunderlich and Lander

Detailed responses to reviews (Jan 8, 2023)

We thank the reviewers for their extensive, thoughtful comments on this manuscript. We have made corrections, carried out all the requested additional experiments, and responded to all of the reviewer's questions. Several figures were modified by the inclusion of additional data, and nine new supplemental figures were added. Through these efforts we believe we have addressed all the issues that were raised. Below, we go through them point by point. Where responses involved incorporating new data, figures, or tables into the manuscript we so indicate.

Two of the reviewers asked that we extend our observations on Myc-induced cell competition as much as possible to the setting of Minute-induced cell competition, and we have now done that, through the addition of multiple new experiments. These are discussed in the individual responses below, and also summarized in a table following the response to Reviewer 2, minor concern 2.

Before going into the details of each point, we also want to say a few words about the interpretation of Caspase-3 staining, since questions about it show up in multiple reviewer comments. Activated caspase-3 staining is a very useful tool for visualizing dying cells, but several factors confound its use for making quantitative comparisons. Four of these are: (1) the number of Caspase-3+ cells in a disc reflects not just the rate at which cells die but also the rate at which dead or dying cells are removed. In experiments in which hemocytes were removed, the decrease in phagocytosis necessarily increases the number of Caspase-3+ cells, confounding the ability to use Caspase-3 staining as a pure measure of cell death; (2) In scenarios in which cell death occurs preferentially at clonal boundaries (as in cell competition experiments), the expected amount of death per disc will depend on the total length of clonal boundary, which may not remain constant across experimental conditions. For example, in an experimental condition that reduced the sizes of clones one might see fewer Caspase-3+ cells per disc, even if the rate of death at clonal boundaries were unchanged, but one might just as easily see more Caspase-3+ cells per disc, if clones in the comparison condition had grown large enough to fuse into large regions without internal boundaries; (3) The assumption that all Caspase-3+ cells are committed to die is not strictly correct, as there is increasing evidence that cells (including *Drosophila* wing disc cells) that activate Caspase-3 can recover and continue to survive and proliferate. Whether this might vary across the conditions studied here is unknown; and (4) It has been shown Myc overexpression in wing disc cells can lead to cell-autonomous Caspase-3 activation (Montero et al., 2008 doi:10.1002/dvg.20373), confounding the ability to interpret all Caspase activation as being a consequence of cell competition.

It was with these complications and caveats in mind that we originally refrained from trying to draw too many strong conclusions from Caspase-3 staining experiments, presenting just a few experiments and placing them at the end of the results section. In response to reviewer comments, however, we've now expanded on these results considerably, with additional explanation and experimentation (Figures 7, S12, S13, S18, S19 and S29, and responses below).

Response to Reviewer #1

(R1 Major Concern 1)

The authors state that 'These results suggest that tumor surveillance and cell competition are, in fact, a single phenomenon' (line 78-79). This is not supported by the data. In fact, types of cell competition have various, stark differences to tumor surveillance.

We softened the indicated statement to “These results suggest that tumor surveillance and cell competition are more closely related phenomena than previously thought”, which we do believe is supported by the data, as described in part in the responses below:

The authors do not present data that is inconsistent with their model, and do not acknowledge published data that predates that model. The authors cite a study from the Vidal lab concerning tumor surveillance, but do not mention major differences in terms of hemocyte involvement: 1) direct attachment of hemocytes to dying cells, 2) juxtacrine Eiger release rather than the proposed disc-wide release, and critically 3) the absence of juxtaposed WT and mutant cells. This causes a small decrease in tumor size but at a late stage of tumorigenesis that cannot really be considered tumor surveillance (Reference 33).

We thank the reviewer for pointing out the issues with citing the 2014 Vidal paper (Parisi et al., 2014) as an example of tumor surveillance. The Parisi et al., paper actually refers to a situation in which all the cells of the disc are transformed into tumors that, while attacked by the immune system, are not eliminated; this is not truly tumor surveillance in the sense we are using here (where initiating tumors are detected and eliminated), and so we have removed this citation (we have also clarified, in the Introduction, that we do not include such situations in our definition of “tumor surveillance”). The Cordero et al., 2010 paper, which we also cited, however, does make the point that “hemocyte-derived Eiger is required for tumor surveillance (it looks at *Igl* clones and shows both hemocyte recruitment and a requirement for hemocyte-derived Eiger for clone elimination), so we have retained this citation.

The authors do not cite a more appropriate study last year from the Bilder lab which fits better with tumor surveillance and cell competition: it studied small oncogenic clones juxtaposed to WT cells that are eliminated before they can form a tumor mass. This is mediated by circulating Eiger from the fat body but does not involve hemocyte Eiger. Overall the manuscript would benefit from clear distinctions between different types of cell competition.

The recent paper from the Bilder lab (de Vreede et al, 2022), which we now cite, is intriguing. Much of it is about the binding of an Eiger-Venus fusion protein to its receptor and draws conclusions about which we have some concerns—these are discussed in connection with a later comment (R1 Concern 5—see below).

Here we focus on the functional experiments in the de Vreede et al., study. They use RNAi to knock down Eiger in various tissues, including hemocytes, the disc and the fat body, to assess effects on survival, growth, or cell death of *scrib* or *dlg* clones or stripes. Their conclusion that hemocyte-derived Eiger is unimportant disagrees strongly with the results of Cordero et al., 2010 who, using *Igl* clones, found a clear requirement for hemocyte-derived Eiger in clone elimination. It also disagrees with the work of Parisi et al. who, as mentioned above, established a critical role for hemocyte-derived Eiger in the response to growing tumors.

Some unusual methodological features of the de Vreede et al., study merit pointing out. First, most of the functional experiments use a novel system in which a temperature controlled Gal4 is used to inducibly knockdown *dlg* in the *ptc*-expressing stripe at the anteroposterior boundary of the wing disc (whether this is truly a model of tumor surveillance is, in fact, debatable—the *ptc* stripe creates a large domain of contiguous tumor cells that does not actually get eliminated by the organism’s response). Since a Gal4 driver is also used in the same flies to knock down Eiger in hemocytes, that knockdown is unavoidably induced at the same time. Given that Eiger/TNF is held in storage granules for subsequently release, it is unclear how much time must elapse before knockdown of gene expression would lead to loss of functional Eiger, so it is unclear if Eiger was turned off early enough. Moreover, it is not demonstrated in these experiments how much Eiger gene expression is knocked down, and whether that would have been adequate to block an Eiger-dependent function (i.e. there is no positive control for the RNAi, which is important given that there is a Gal80 in the background, which may reduce Gal4 function even at the permissive

temperature). Moreover, the use of the two Gal4 drivers together ensures that whenever they knockdown Eiger in hemocytes (or the fat body), *dlg* will also be knocked down in the same tissues, the consequences of which, if any, are unknown (they do show that knocking down *dlg* in in hemocytes or the fat body doesn't exacerbate cell death caused by knocking down *dlg* in the *ptc* domain, but there was no reason to believe that it should have had that particular effect).

Given these concerns, de Vreede et al. also carried out a small number of more conventional experiments in which *scrib* twin-spot clones were generated, and Gal4 was used solely for Eiger knockdown. In this case, although they observed that hemocyte specific Eiger knockdown was not sufficient to block clone elimination, the key experiment of showing that it was not necessary appears not to have been done. Specifically, fat body-specific knockdown was only examined in combination with hemocyte specific knockdown. We are not sure why the experiments were done this way, but it suggests fat body-derived Eiger may be not be fully sufficient for tumor surveillance (which would not be inconsistent with our findings, as we do not see *complete* elimination of cell competition with hemocyte knockdown of Eiger). It appears that the only other situation in which de Vreede et al., showed an effect of Eiger knockdown solely in the fat body is in the context of the response to cell wounding; in that setting, however, hemocyte knockdown of Eiger had exactly the same effect as fat body knockdown (which, incidentally, is inconsistent with their model).

Ultimately, while the model proposed by de Vreede et al. is elegant, we suspect it is likely incomplete. We find it both curious and intriguing that investigators working within both the tumor surveillance and cell competition areas seem so often to reach discrepant conclusions regarding whether Eiger from epithelial cells, hemocytes, or the rest of the organism is most important. As we suggest in the Discussion section (and have now amplified on further), all three sources may matter, but due to the existence of positive feedback loops, the matter of which sources are necessary in specific contexts may be very dependent on subtleties of experimental details. We think the added discussion improves the manuscript and thank the reviewer for calling this issue to our attention.

(R1 Major Concern 2)

The authors use a Myc cell competition model for most experiments but only check a few with a Minute cell competition model as well (Figure 1, Figure S6). I believe that key experiments should be addressed with Minute cells to justify a generalized claim, as the title of the manuscript indicates. Please test whether hemocyte-produced Eiger is important for Minute cell competition...

As shown in new Figure S20, knockdown of Eiger in hemocytes does reduce cell competition in the *Minute* assay, to the same extent as hemocyte ablation.

...[please test whether] cell death in Minute clones is reduced once hemocytes are ablated.

As shown in new Figure S20, in *Minute* experiments, Caspase-3-positive cell numbers decreased significantly following Eiger knockdown in hemocytes, although the effect was partial, and in response to hemocyte ablation the difference was highly variable and not statistically significant (see next response for further explanation).

*...[The] Moreno lab previously showed that hemocyte ablation by *He>hid* increased, not decreased, the number of C3 positive cells in Minute disc (Reference 30), which would be the opposite of the author's model.*

As noted at the beginning of this letter, interpreting Caspase-3 staining quantitatively when exploring cell competition is complicated by many factors, not the least of which is that the number of stained cells reflects both the rate at which cells are killed and the rate at which they are removed by extrusion and/or phagocytosis. In the case of the Moreno lab's paper (Lolo et al., 2012), the increase in Caspase-3-positivity in hemocyte-deficient discs was attributed to a loss of

phagocytosis. There are several reasons why we would not necessarily have expected to obtain the same result as they did.

First, in our studies, cell competition was significantly blocked by hemocyte ablation. Lolo et al., don't actually report whether cell competition was inhibited by hemocyte ablation, and it is difficult to tell by examining their published figures. One clear difference between our studies and theirs is that they report that hemocyte ablation (using hemese-Gal4 to drive UAS-hid) produced only an 80% reduction in hemocyte levels, whereas in our experiments we used hml-Gal4, and appear to have achieved much more complete ablation (see Figure S3, and discussion below in response to R1 Major Concern 3). If, due to incomplete ablation, they failed to inhibit cell competition, but still did a good job of inhibiting the removal of cell corpses, then you would expect them to observe an increase in total Caspase-3+ cells in hemocyte-ablated discs. In contrast, in our case, with more complete ablation and, concomitantly, significant inhibition of cell competition, you'd expect fewer Caspase-3+ cells to be generated in the first place. With both cell death and cell removal going down, you might expect the overall change in Caspase-3 staining to be intermediate, which is in fact what we see. Particularly with *Minute* cell competition, the change in total Caspase-3+ cells with hemocyte ablation is insignificant (Fig. S20, as mentioned above). In this context, the fact that the decrease in Caspase-3+ cells becomes significant when using RNAi to knockdown Eiger in hemocytes (Fig. S20) also makes sense—Eiger knockdown would not be expected to block phagocytosis, so in that experiment we would be just looking at the effects on cell death.

In addition to the possibility stated above, there are several other differences in methodology between our study and the Lolo et al. paper that could potentially account for their results being much more sensitive to changes in phagocytotic rate than ours. For example, they counted both dead cells and extruded "fragments", whereas we gated on Caspase-3+ spots of sufficient size to represent cells. With a single dead cell potentially producing many fragments, this could strongly weight the observations of Lolo et al., toward extruded, unphagocytosed corpses. In addition, we routinely washed discs to remove non-specifically bound hemocytes (we used the same washing protocol in all experiments even if hemocytes were not going to be visualized), and this might have selectively removed basally-attached cell fragments. Another factor is that they assessed their results at 72 hours, as opposed to 96 hours in our case. It is possible that, with the additional time, hemocyte-independent mechanisms of cell removal (e.g. Li and Baker, 2007 and Ohsawa et al., 2011) might have been able to clean up much of the apoptotic debris.

...The authors could tone-down the generality and use the term Myc cell competition more specifically.

We did note, particularly after having increased the number of *Minute* experiments in this study, that the effect of hemocyte ablation on *Minute* cell competition is generally not as great as it is on *Myc* cell competition. This suggests that disc-intrinsic or systemic mechanisms may play a greater role in the former case than the latter. While we don't believe this seriously impacts the generality of our conclusions, we do now discuss this difference in the manuscript.

(R1 Major Concern 3) The authors use a combination of HmlΔ-Gal4 and UAS-hid to ablate hemocytes. However, a study from the Theopold lab shows that expression of apoptotic inducers by HmlΔ-Gal4 may increase overall hemocyte numbers that are not hml-positive (Arefin et al., PloS one, 2015, doi:10.1371/journal.pone.0136593). The authors should show the efficiency of hemocyte ablation using established hemocyte markers such as srp, Hemese, or NimC1. Furthermore, could hemocyte ablation have secondary effects, e.g. on antimicrobial peptide production as seen in the Theopold lab paper, which is known to regulate cell competition?

We carried out the requested experiment in which we used *srpHemo-H2A-3xmCherry* to label hemocytes in *hml>hid* larvae. We imaged discs in both 'No Wash' and regular conditions, and

observed no mCherry-positive cells in the wing discs subjected to hemocyte ablation (new Figure S3).

Of course, as the total number of hemocytes in a wing disc is relatively small (even under no-wash conditions) this negative result cannot rule out the possibility that a small number of *Hml*-negative hemocytes do remain and, indeed, the population that Arefin et al. report as increased is the lamellocytes, which are a minor population (1-5%). So formally, we cannot rule out the possibility that the increase in lamellocytes that Arefin et al., reported plays some role in blocking cell competition. However, if that were the case it would not explain why specifically knocking down Eiger using *hml>Gal4*—which would spare lamellocytes—has the same effect as hemocyte ablation. In view of this we think the explanation that hemocyte ablation works via increasing lamellocyte numbers is unlikely, even if still formally tenable.

(R1 Major Concern 4) *...the data are not fully convincing that cell competition sites recruit hemocytes and that disc-attached hemocytes are functionally important. This is based on these three observations from the manuscript:*

1) In 'No Wash' condition, the number of disc-attached hemocytes seem to be unchanged between control and Myc conditions (Fig. S7Q), which does not seem consistent with the author's argument that hemocytes are associated specifically with discs undergoing cell competition...

We're sorry we didn't make the point of this experiment clearer. The vast majority of hemocytes observed in the No Wash condition are loosely and—we'd argue—non-specifically bound to discs (which, after all, are bathed in hemolymph). It is therefore not surprising that, without washing, one observes about the same high number of them in all discs. It is only after a sufficient number of gentle washes that the difference between control and Myc conditions emerges, as would be expected following the removal of a large pool of nonspecifically bound cells. The other point made by Fig. S10 and Fig. S11 (current numbering) is that too aggressive washing can remove all the hemocytes from any disc.

...2) Hemocytes appear to accumulate on a defined locus (anterior-, ventral-facing portion of the disc), which is inconsistent with the idea that hemocytes are recruited to sites of cell competition.

We actually do not argue that hemocytes are *initially* recruited to sites of cell competition, but rather that they are first recruited to discs in which cell competition is taking place, and only after about 72 hours do they come to be significantly overrepresented near clonal boundaries (Fig. 2E). In any case, to address the reviewer's specific question, we agree that hemocytes tend to show up preferentially in an anteroventral position; this corresponds to the location of a pronounced fold in the wing disc. Our interpretation is the hemocytes trapped in folds are more protected than hemocytes elsewhere from being dislodged, both by the vigorous contractile movements of the larva as it feeds, and by the washing of discs after they have been dissected. As is clear from Fig. S10, even specifically-bound hemocytes can eventually be removed by enough washing, and we suspect that what we routinely observe after 4-5 washes has withstood not just the removal of non-specifically bound hemocytes but also had some specifically-bound ones as well. Since hemocytes in folds would be expected to withstand washing better, we are not surprised ultimately to find preferential retention of them there.

3) Hemocyte attachment on Myc mosaic discs is very variable and some Myc discs have no hemocytes attached (Fig. S7Q). However, hemocyte ablation very consistently results in a lack of competition. Together this implies that the attachment of hemocytes onto discs may not be necessary for hemocytes to act on discs that undergo Myc cell competition.

As stated in the previous response, what we routinely observe after 4-5 washes probably reflects the removal of some specifically bound hemocytes, so we cannot be confident that a disc in which we didn't observe hemocytes lacked any specifically bound ones at the time of dissection. In addition, hemocytes are highly motile cells and likely move on and off discs throughout the

experimental period. It seems likely that the cells we observe at the time a disc is dissected represent only a fraction of those that have interacted with the disc.

(R1 Major Concern 5) 1) What is the role of hemocyte-originated Eiger and where does the TNF ligand bind in Myc cell competition? Drosophila has two TNFR receptors, Wengen and Grindelwald. At least for Grindelwald, the Bilder lab showed that the receptor is exclusively located on the apical side of cells in the epithelial wing disc tissue (de Vreede et al., Science, 2022, doi: 10.1126/science.abl4213). Therefore, it seems that the receptor would not be able to directly bind the TNF ligand from hemocytes in proximity, unless the disc was disrupted somehow..

We did not address which of the two Eiger receptors, Wengen or Grindlewald (*grnd*), serves as the receptor for hemocyte-derived Eiger. Both are expressed in wing discs, although data from de Vreede et al, and others (e.g. Kodra et al., 2021, bioRxiv) suggest *grnd* is required for cell competition.

We do not think the data in the de Vreede et al. paper rule out the possibility of *grnd* serving as a receptor for basolaterally-supplied hemocyte-derived Eiger. While their results suggest that, at steady state, a very large amount of *grnd* is located apically, they do not show that all *grnd* is localized there, and one can clearly see in their images small to moderate amounts of *grnd* staining at the basolateral surface. Indeed, they even report that Eiger-Venus binds to basolateral surfaces, but apparently do not test whether that binding is *grnd* dependent.

It is not unusual for receptors to cycle between apical and basal compartments in epithelial cells, such that steady state distribution is not a good predictor of signaling capacity. While de Vreede do provide evidence that polarity mutations enable greater access for Eiger to an apical receptor pool, they *do not* establish that such additional binding is necessary for Eiger function in such cases. In the end, their model for how polarity-deficient clones are eliminated is an elegant one, and they present data consistent with their model, but we believe there are many aspects of the proposed cell biology that have yet to be proved. In addition, we hasten to add that the receptor-binding characteristics of the Eiger-Venus fusion protein used by de Vreede et al., are apparently unknown (adding a protein tag frequently alters binding affinity and specificity); in fact, we have been unable to find evidence in the literature that this fusion protein is functional.

2) The authors show a complete clone size rescue for Myc cell competition when hemocytes are ablated (Fig. 1). However, the effect on cell death in this condition is minor (Fig. 7). How do the authors reconcile this?

As mentioned at the start of this letter, using Caspase 3 staining as a quantitative measure of cell death is complicated. For example, in this case, we should expect the effect of hemocyte ablation on cell death to be offset, at least partially, by a decrease in the phagocytosis of dead cells (leaving more Caspase 3+ “corpses” around to be stained). Thus we would not expect to see as great a rescue of Caspase 3 staining in Fig. 7 as we see for clone size in Fig. 1. We now discuss this point in the manuscript.

In addition, there are also possible reasons why hemocyte ablation might actually have a stronger effect on clone size than on cell death. For example, hemocytes carry out phagocytosis, and it has been argued that phagocytosis itself is required for cell competition (Li and Baker, 2007). Thus, even if loss of hemocytes only partially blocked cell death it would not be unreasonable to expect to see a complete block of competition (as measured by clone sizes). Another possibility is that hemocytes might play a role in ensuring that the proper cells die, rather than just whether cells die.

They could analyze other factors that may be changed by an ablation of hemocytes in Myc cell competition, such as growth rate or JNK activation changed in loser cells.

As requested, we analyzed JNK activation during Myc cell competition, using anti-phospho-JNK antibodies. The levels of JNK activation are elevated at the sites of cell competition, and this

increase is suppressed when hemocytes are ablated (See new Figure S15). We thank the reviewer for suggesting this experiment.

(R1 Minor Concern 1) *In the title, it is unclear what ‘directed’ is supposed to mean. The model I think proposes that hemocytes are more permissive than instructive for competition. Please revise for a more appropriate description.*

We have replaced ‘directed’ with ‘promoted’.

(R1 Minor Concern 2) *Add clear labeling to all figures, e.g. Fig. 7 misses Cas3, GFP, DAPI labeling.*

We have added clear labeling to Figure 7 per the reviewer’s suggestion.

(R1 Minor Concern 3) *Is hemocytes adherence to eye imaginal discs similar to wings, in that the difference is seen only with an intermediate number of washes? Please clarify.*

Yes, the situation is similar in eye discs. To compare with the data in Figure 3, we gathered additional eye disc data from “no-wash” and “thoroughly washed” conditions, and have added them as new Fig. S11. As with wing discs, we observe a large number of attached hemocytes prior to washing, regardless of genotype, and with extensive washing we can remove all hemocytes, regardless of genotype.

(R1 Minor Concern 4) *Please clarify the statistical analysis used. What do they solid bars and whiskers on the graphs mean? Add a supplemental table with all p-values and statistical tests listed as well as a proper labeling of graph error bars in the figure legends. Use sufficient numbers of samples for experiments– some experiments, as for example Fig. 7A and Fig. 7C, use only n=5. This is unfortunately not sufficient to make a solid conclusion.*

In all box-whisker plots, the lower whisker represents the minimum data value, and the upper whisker the maximum. The white bar traversing each box is the median, and the box limits give the boundaries of the upper and lower quartiles (i.e. the middle 50% of the data lie within the box). As most of the data were not well fit by a Normal distribution, we used the Mann-Whitney U test, a nonparametric test, to compare outcomes between independent groups. We also used the Chi-squared test to identify whether Caspase-3-positive cells were equally distributed in *Myc* flip-out clones and regions surrounding these clones with the same sizes (new Figure S13K, original Figure S9). We have added a supplemental table with all p-values and statistical tests listed (Table S2). Although the statistical testing does take the number of samples into account, we agree it is important to provide more samples than five for individual data points. There were already more than 5 discs in Figure 7A and Figure 7C, so we assume the reviewer here meant Figure 6A and Figure 6C. We now have added more data points to Fig. 6, making the significant differences more apparent (see updated Figure 6).

(R1 Minor Concern 5) *Does the number of Caspase3-positive cells change in *Myc; Hml>egr RNAi* and *Myc; Hml>bskDN*?*

We did the requested experiments; please see the response to Reviewer #2 Major Concern 8 and Reviewer #2 Major Concern 10.

(R1 Minor Concern 6) *line 42: The sentence is incorrect. The Pastor-Pareja lab used *dlg* mutant discs that do not show cells in juxtaposition (Ref. 27).*

We thank the reviewer for catching this; we inadvertently cited the wrong paper. We have now corrected this in the manuscript

(R1 Minor Concern 7) *line 89-107: Please cite any previous papers that developed these systems and showed that they induce cell competition; if this approach is original, perhaps state*

why the system used here is different than previous work. It would help a lot to add a Supplementary Figure that diagrams the clonal induction: which clones are generated in Myc and Minute cell competition models and how are they marked (GFP+, GFP-, lacZ+, lacZ-).

In this paper, we used three methods to generate clones. We used *FRT82B,tub-HA:Myc* to generate *Myc* 'twin-spot' clones inducing cell competition as described by Steiger et al. We generated *Minute* 'twin-spot' clones as Li and Baker did, but using *FRT40A* instead of *FRT42*. Moreno and Basler previously induced cell competition by generating *wild-type* flip-out clones in a *dMyc* background. Similar to that, but with a minor change, we generated *dMyc* clones in a *wild-type* background. We did observe more Caspase-3-positive cells located in regions surrounding clones, indicating that cell competition was induced (new Figure S13K, original Figure S9).

A supplemental figure has been added, Figure S1, that diagrams these three scenarios, and the Methods section has been appropriately augmented.

(R1 Minor Concern 8) *line 138: Why is it notable that hemocytes recruitment is 'early'? This should be a time when cells are being eliminated and expressing Caspase 3.*

The reviewer is correct. We removed the phrase in question

(R1 Minor Concern 9) *Fig. 4 shows the same experiment with the reporter either inserted on 2nd or 3rd chromosome. Please move one of the experiments to the Supplementary Figures.*

We moved one of the experiments to the Supplementary Figures (Please see updated Figure 4 and new Figure S14).

(R1 Minor Concern 10) *line 223: Please explain the discrepancy with the study from the Johnston lab (ref. 59).*

Moreno et al., (2002) reported that, during *Minute*-induced cell competition, loser cells are eliminated by JNK-dependent apoptosis due to reduced Dpp signaling during *Minute*-induced cell competition, and it appears that the majority of researchers in the field have accepted this conclusion as valid. However, a recent study from the Johnston lab argued that JNK is *not* required for removal of wild-type loser cells during *Myc*-induced cell competition. Unfortunately, the Johnston lab study (Kodra et al., 2021) exists only as a post to bioRxiv, and after 3.5 years has yet to be published as a peer-reviewed article. We cited that work out of respect for the high quality of the studies that lab has produced, but given the un-reviewed nature of the findings we thought an extended discussion would be ill-advised.

(R1 Minor Concern 11) *Fig. 7: Is apoptosis restricted to the low Myc-expressing cells? Is apoptosis localized at the clone boundary mainly?*

Please see response to Reviewer #2 Major Concern 11.

(R1 Minor Concern 12) *Fig. S3E: Why is the shown disc either earlier in development or at a lower magnification?*

Please see response to Reviewer #2 Major Concern 3

(R1 Minor Concern 13) *Fig. 1F: Why are some lacZ+ cells less bright and hazy than others in Minute; hemocyte-ablated disc? It seems that there are two classes of cells.*

We thank the reviewer for pointing this out. In this panel we inadvertently chose the level of optical slice too close to the apical surface and ended up including some clones of peripodial cells (which are shaped very differently than columnar cells and stain with a different intensity). We have picked a more basal slice in the same disc and updated Figure 1.

(R1 Minor Concern 14) *Please specifically address why the authors' results conflict with those of the Moreno lab (ref. 30) for hemocyte requirement on death of cells.*

Although the paper by Lolo et al., 2012 (formerly ref. 30) addresses the role of hemocytes in cell engulfment during cell competition, there are actually very few experiments in that study that speak to the issue of whether hemocyte ablation affects cell competition itself. Their Fig. 6 is the only place they visualize cell competition clones in combination with hemocyte ablation, and only a single disc is shown, from which it is difficult to say whether there has been a significant effect on cell competition (they do not comment on this issue in the manuscript either).

As mentioned in the response to Major Concern 2 (above), a significant difference between the Lolo et al. study and ours is that they used *He>Gal4*, rather than *Hml>Gal4* to drive hemocyte ablation, and they report being able to ablate only 80% of hemocytes. Given that, for Minute cell competition, inhibition due to hemocyte ablation was, in our hands, only partial, it would not be surprising if, in the work of Lolo et al., there had been no noticeable change in cell competition.

Accordingly, we don't see any real conflict between the Lolo study and ours regarding the role of hemocytes in the death of cells. And in other respects, such as the recruitment of hemocytes by cell competition, the study agrees well with ours.

Please also see response above to Major Concern 2, which specifically discusses the Caspase staining results from the Lolo et al., paper, and their relationship to our work.

(R1 suggested Enhancements): *How are hemocytes attracted to the tissue undergoing cell competition, if at all? The authors show in Fig. 4 that hemocytes are attracted through cell juxtapositions. In the discussion (line 314-316), the authors mention that ROS or basement membrane degradation may lead to the attraction. They could address whether a correlation between ROS or basement membrane degradation and hemocyte attachment is present in Myc cell competition.*

We appreciate the suggestions, and this is certainly an area in which we are currently working. However, we feel that there aren't any great experiments that we can do at this time to address these questions in a sufficiently definitive way. For example, while methods do exist for detecting ROS responses in cells, we would not necessarily expect a correlation in localization with ROS signals to be any stronger than the significant but not so strong correlation we currently see with the locations of clonal boundaries. Similarly, while it might be interesting to see whether sites of hemocyte attachment correlate with sites of basement membrane degradation, experiments would be needed to determine whether such degradation was likely to be a cause, rather than a consequence, of hemocyte attachment. As a result, we believe such experiments will need to be done as part of future work.

Response to Reviewer #2

(R2 Major Concern 1) *In the experiments depicted in Figure 1, how long was the egg laying of the crosses? Can the authors provide more information regarding the protocol for the generation of clones at different time points? (Egg laying, heat shock time, and heat shock duration). Do the authors find any developmental delay when they ablated hemocytes? For example, in Figure 1E the disc looks younger than in 1F. This is more important for Minute competition, where the twin spot (Minute/Minute) dies therefore there is no clear indication of the recombination event. For example, the difference in the area of the "dark clone" in Figure S2 could be due to the different developmental stages that the heat shock was applied.*

For *dMyc* twin-spot larvae, eggs were collected for 1 hour, then kept at 25°C for 3 days followed by a 1-hour 37°C heat shock and 72 more hours at 25°C before dissection. For *Minute* larvae, eggs were collected for 1 hour, then kept at 25°C for 2 days followed by a 1-hour 37°C heat shock and 96 more hours at 25°C before dissection. We have added this more detailed protocol to the method section.

We have also performed an egg-laying assay on *wild-type* and *hml>hid* larvae. We treated their parents, which were previously kept at 25°C, with CO₂ before letting them lay eggs for 1 h

and discarding those eggs. Then we collected eggs for 1 h and kept them at 25°C. After that, we dissected larvae at different time points---84, 96, 108, 120, or 132 h after egg laying. We stained the wing discs with anti-Ptc and anti-Wg antibodies and quantified the posterior compartment sizes of them. No obvious difference can be observed between these two genotypes among all these time points, so we see no evidence of developmental delay (new Figure S4).

(R2 Major Concern 2) *Have the authors stained for apoptotic markers in genotypes of Figure 1 to check if apoptosis of Minute loser cells is reduced upon hemocyte depletion?*

This is addressed in the response to Reviewer #1 Major Concern 2.

(R2 Major Concern 3) *In Figure S3 the discs have different sizes. Is that due to differences in imaging or this implies differences due to developmental stages? If this is the case, the question that comes to my mind is if there is any difference in hemocyte attachment at different stages since the folding of the wing discs is increasing in the late 3rd instar larva and the authors state that they have observed accumulation of hemocytes associated with folds (line 133)*

Wild-type wing discs have very few attached hemocytes (many have none at all); in the interests of being conservative in our choices of what images to show, we picked a wildtype disc at the upper end of hemocyte numbers. Unfortunately, that happened to be a relatively small disc, not typical of the entire set. We have replaced that image with a more representative one(see new Figure S6, original Figure S3). We thank the reviewer for bringing this to our attention.

(R2 Major Concern 4) *In Figure S7, where the authors support that washes affect the hemocyte attachment, I am concerned since the difference is obvious in 4 washes, while it is not obvious (not statistically significant) in 2, 3, or 5 washes. If each dot depicts one disc, in 5 washes for example 4 discs of control behave as 3 discs of myc discs and there are only 2 discs that show an increased number of attached hemocytes. Here, my concern is also related to the developmental stages of the discs and how this could affect the attachment of the hemocytes. If each dot is one disc, I recommend that the number of discs in each wash should be increased. Especially since this result is contracting an earlier study (#58 Reference) and also supports that hemocyte attachment is affected during competition.*

We have increased the number of discs in each wash, as requested. Although the differences were statistically significant previously for both 4 and 5 washes, those differences are now more visually obvious (see new Figure S10, originally Figure S7).

(R2 Major Concern 5) *In Figure S8B the ratio dark/bright is higher in the eye imaginal disc, which does not occur in S8C when p35 is expressed by GMR driver. Have the authors checked whether in eye discs their Myc super competition system supports the death of losers when are in close proximity to 2xmyc clones?*

We have stained eye discs carrying neutral or *dMyc* clones with anti-Caspase-3 antibody. The number of Caspase-3-positive cells was increased significantly in *dMyc* discs, both when the total number of Caspase-3-positive cells in the whole eye disc were compared and when only the number of Caspase-3-positive cells located at clonal boundaries were compared (see new Figure S12, original Figure S8).

GMR is expressed posterior to morphogenetic furrow (MF) (therefore mainly in post-mitotic cells), but not in the anterior to MF proliferative cells. How the competition anterior to MF is rescued by GMR-p35 since the competition in proliferative cells will still impact the final sizes of the clones? Have the author checked the hemocyte recruitment and the clonal areas in the eye discs in genotypes shown in Figure 3D-F, where apoptosis is blocked by hid1 heterozygosity?

We thank the reviewer for pointing out our error. It is true that GMR is primarily expressed posterior to the MF, however after the MF passes one further round of cell division, the second

mitotic wave (SMW), occurs, so there is an opportunity for competition posterior to MF to be rescued by *GMR-p35*. The reviewer is correct, however, that we should have excluded measurements anterior to the MF in the *GMR-p35* experiments, which we have now done in the revised Figure 3. Doing that improved the data—the initial clone size ratio is larger so the observed effect of *GMR-p35* is a greater reduction than before (see updated Figure 3).

(R2 Major Concern 6) *In Figure S9 the authors argue that cell competition happens when dMyc flip-out clones are generated in a wild-type background. Please provide an apoptotic marker for these genotypes, to show that cell death is induced outside the myc overexpressing clones. This is necessary since their conclusion for hemocyte recruitment due to clonal juxtapositions depends on proving that this system behaves as a supercompetitor system.*

We have assessed apoptosis by staining these wing discs with anti-Caspase-3 antibody. We counted the number of Caspase-3-positive cells within *dMyc* flip-out clones and equal areas around each clone in each disc respectively. We found that Caspase-3-positive cells were preferentially located outside clones (see new Figure S13).

(R2 Major Concern 7) *Have the authors checked the hemocyte recruitment when they have two copies of tub-HA:Mycw+ transgene (genotype: FRT82B, tub-HA:Mycw+/ FRT82B, tub-HA:Mycw+). I believe this will strengthen their conclusion that “it is not dMyc overexpression per se, but rather the juxtaposition of cells with different levels of dMyc, that drives hemocyte recruitment”.*

We have now counted hemocytes in wing discs with two copies of *tub-HA:Mycw+* transgene (genotype: FRT82B, tub-HA:Mycw+/ FRT82B, tub-HA:Mycw+) (see updated Figure 4). The number of hemocytes is not increased in these discs, consistent with our conclusion.

(R2 Major Concern 8) *The authors in Figure 5 present a requirement of egr by hemocytes for myc-dependent cell competition, by measuring the growth of clones during myc super competition. Could the authors provide the caspase-3 staining in those genotypes presented in Figure 5.*

We stained for Caspase-3 activation in wing discs with the genotypes of Figure 5. Consistent with our previous results, the number of Caspase-3-positive cells in wing discs undergoing *dMyc*-induced cell competition was increased, and this number went down significantly when either of the two *egr-RNAi*s was expressed only in hemocytes, using *hml-Gal4* (see new Figure S18).

(R2 Major Concern 9) *One also important conclusion in the paper is shown in Figure 6, where hml>bskRNAi and hml>bskN is sufficient to block myc-dependent competition (estimated by area ratio of the clones). Please provide representative images of these experiments.*

We had provided these images in original Figure S10, which is now Figure S16.

(R2 Major Concern 10) *The effect of JNK activation in hemocytes during myc-dependent competition will be strengthened by checking apoptosis in the genotypes of Figure 6. Could the authors rescue the myc-dependent apoptosis when they deplete bsk or overexpress bskDN in hemocytes (by hml driver)? Similar to what they support in Figure 7, where they kill the hemocytes.*

We stained for Caspase-3 activation in wing discs with the genotypes of Figure 6. The number of active Caspase-3-positive cells was reduced significantly when *bsk* was deleted or *bskRNAi/bskDN* was expressed specifically in hemocytes (see new Figure S19).

(R2 Major Concern 11) *In Figure 7 panel F, the black clones have 2 extra copies of Myc and act as supercompetitors. Please provide a “zoom-in” of panel F, since the disc that is shown gives the impression that a significant part of apoptosis is localized in the black areas.*

Because the images in Fig. 7 are maximum intensity projections it is difficult to see boundaries crisply, so we added a panel showing a single Z-slice from a disc with Myc clones (panel G”), as

well as a zoom-in of one area on that disc (panel G'''). It is true that Caspase 3 staining is not restricted to the low *Myc*-expressing cells. We see this consistently. It has been reported that overexpression of *dMyc* in imaginal disc cells can cell-autonomously trigger cell death (Montero et al., 2008, doi: 10.1002/dvg.20373), so this may be what we are observing. We still do see a preference for Caspase 3 staining near clonal boundaries. This can be observed to some degree in the images in Fig. 7 and is more thoroughly quantified in Figure S13 (in which Flp-out *Myc* clones were used, instead of twin-spot clones).

Also, the cell death localization of double GFP clones (losers, with no extra Myc) in the anterior compartment looks quite away of the boundaries.

The anterior compartment staining to which the reviewer refers is actually close to boundaries between double GFP (wildtype) and single GFP (1x *Myc*) clones.

(R2 Major Concern 12) *In Figure 7B the cell death is quite a lot. Have the authors checked the cell death in the genotype of 7B without heat shock (hsFLP; FRT82B, ubiGFP/FRT82B, tub-HA:MycW+)?*

Previously, we used a more generous threshold for counting Caspase-3-positive cells, which we believe led to inclusion of a significant amount of background staining, and a larger proportion of small, puncta arising from cell fragments. We went back and re-counted all experiments using a more stringent threshold, and although no conclusions were altered, we find that the Caspase-3-positive cell counts are now more in line with those reported in other studies.

We also did carry out anti-Caspase-3 staining of wing discs with the genotype of 7B without heat shock (*hsFLP; FRT82B, ubiGFP/FRT82B, tub-HA:MycW+*), as requested. As shown in the updated figure 7, we observed no significant difference between the numbers of Caspase-3-positive cells in these discs and wing discs carrying control clones.

(R2 Minor Concern 1) *In Figure 1D, the twin spot clones have two copies of arm-lacZ and the background has only one. It is not obvious in this image the difference in the background with the twin spots (1 vs 2 copies arm-lacZ). How long was the heat shock applied in such experiments?*

We have adjusted the contrast (see updated Figure 1), which we believe makes the differences stand out more clearly. The heat shock was applied for 1 h in such experiments.

(R2 Minor Concern 2) *Please state clearly the conclusions that cover both the cell competition system (myc or Minute) and the conclusions for only the Myc system. For example, the Minute competition was used only for two conclusions: a) ablation of hemocytes leads to higher occupation of the winner area compared to Minutes, and b) hemocytes are recruited to sites of competition. All the other results have not been shown for the Minute competition. This should be clear in the title of the paper (or in the abstract), in the title of the result section, and in the discussion. E.g. Hemocyte recruitment occurs regardless of whether cells die or competition succeeds in myc super competition, Expression of Eiger/TNF by hemocytes is necessary for myc-dependent cell competition, Recruitment of hemocytes during myc competition does not require JNK signaling, Hemocytes contribute to killing loser cells in Myc super competition. I was kind of puzzled about which conclusions have been shown in Minutes.*

In response to these concerns we have extended many of the observations made using the *Myc* system to the *Minute* system as well. Our new results show that hemocyte-produced Eiger is important for *Minute* cell competition, and cell death in *Minute* clones is reduced when hemocytes Eiger is knocked down. The table below summarizes which experiments, at this point, have been performed in both systems, and which have been done with only *Myc* cell competition. It also points out any salient differences observed between the two systems. Overall, we believe the results support the view that hemocytes play an important role in both types of cell competition, although that role may be somewhat weaker in the case of *Minute* cell competition.

Experiment	Myc	Minute	Salient differences
Hemocyte ablation rescues cell competition.	X	X	The rescue effect of hemocyte ablation on Minute cell competition is not as great as it is on Myc cell competition.
Hemocytes are recruited to sites of cell competition.	X	X	
Hemocyte recruitment occurs regardless of whether cells die or competition succeeds.	X		
Hemocytes are recruited by clonal juxtapositions.	X		
Expression of eiger by hemocytes is necessary for cell competition	X	X	
Recruitment of hemocytes does not require JNK signaling	X		
Hemocytes contribute to killing loser cells partially	X	X	Hemocyte ablation reduces the number of Caspase-3-positive cells for Myc cell competition but a significant effect was not observed with Minute cell competition.

(R2 Minor Concern 3) Please also provide references where the same fly stocks (*FRT82B,tub-HA:Myc*) have been used in super-competition

The three methods used to induce cell competition in this study are now diagrammed and explained in new Figure S1 and the relevant references are made clearer. For supercompetition using twin spot clones, the *FRT82B,tub-HA:Myc* construct was as described in Steiger et al., 2008 (cited).

Response to Reviewer #3

(R3 Comment): Although the hemocytes produce *Eiger/TNF*, it is the same signal as during epithelial cell competition, it is unclear to what extent hemocytes-mediated cell killing contributes to cell competition. Thus, this study does not provide a novel mechanistic sight into cell competition or tumor surveillance.

Taken together with earlier literature, our data suggest that not just epithelial *Eiger*, but both epithelial and hemocyte sources of *Eiger* are needed for cell competition. In the discussion we explain how this might provide discs with an important fail-safe mechanism, allowing for sensitive recognition of meaningful cell-to-cell differences without loss of specificity (i.e., avoiding killing cells that are merely undergoing phenotypic fluctuations). We believe this is indeed a novel mechanistic insight.

(R3 Major Concern 1) Figure S5 shows hemocytes are attracted via cell competition only in the peripheral part of the imaginal disc. How do the authors explain this? What is the signal from the winners/losers attracting hemocytes?

As noted in the response to Review 1 Major Concern 4, hemocytes do tend to show up disproportionately where there are folds in the wing disc. Our interpretation is that hemocytes trapped in folds are more protected than hemocytes elsewhere from being dislodged by washing. As is clear from Fig. S10, even specifically-bound hemocytes can eventually be removed by enough washing, and we suspect that what we routinely observe after 4-5 washes has withstood not just the removal of non-specifically bound hemocytes but also had some specifically-bound

ones as well. Since hemocytes in folds would be expected to withstand washing better, we are not surprised ultimately to find preferential localization of them there.

As for the nature of the signal that attracts hemocytes, we do not know what it is at this point. Orthologues of the major attractants of vertebrate macrophages (chemokines) are not found in the *Drosophila* genome. The Moreno lab has previously reported that loser cells release Tyrosyl-tRNA synthetase fragments which can recruit hemocytes (Casas-Tintó, et al., 2015 doi: 10.1038/ncomms10022), but given the well-stirred environment in which discs reside, we believe that a change in the adhesive properties of the disc itself would be a more efficient way to capture hemocytes than the release of a chemoattractant. Speculation about this process can be found in the manuscript (next to last paragraph of Discussion).

(R3 Major Concern 2) *In Figure 5, is it significantly different between the groups: Myc;hemocyte-ablated vs Myc;egrRNAi? If yes, that suggests another factor potentially mediating this process.*

Yes, it is significantly different between the groups: *Myc;hemocyte-ablated vs Myc;egrRNAi*. There are a few possible explanations. First, RNAi-mediated knockdown is rarely complete, so some Eiger may remain. Second, hemocyte ablation eliminates phagocytosis by hemocytes, whereas elimination of Eiger should not. Third, hemocytes may produce something, in addition to Egr, that contributes to cell competition, i.e. Eiger may be only part of the story.

(R3 Major Concern 3) *In Figure 7, is it significantly different between the groups: control vs Myc;hemocyte-ablated? A simple interpretation for this is that the contribution of hemocytes to Myc-induced cell killing is only in part.*

Yes, it is significantly different between the groups: *control vs Myc;hemocyte-ablated*. We agree that the most parsimonious interpretation of the data is that hemocytes contribute to cell killing but are not responsible for the entirety of it. This agrees with other published studies showing that epithelial cells themselves can kill their neighbors during cell competition (e.g. Li and Baker 2007). We have attempted to make this viewpoint more apparent in the manuscript.

(R3 Major Concern 4) *In [the] summary, the conclusive remark "...tumor surveillance is less a mechanism evolved to attack cancers than a manifestation of a general strategy for ensuring conformity of epithelial cell behaviors." is difficult to understand in relation to this study.*

This is explained in more detail in the last paragraph of the introduction. We have also rewritten the phrase at the end of the summary to try to make the meaning clearer. This point was meant to acknowledge a longstanding conundrum about the evolution of tumor surveillance. With most naturally occurring cancers arising late in life, past the age of reproduction, and in short lived organisms such as *Drosophila* probably never having the time to arise at all (in nature), it is difficult to see why evolution would select for and maintain a process like tumor surveillance. Certainly, one wouldn't immediately expect very strong selection for it, even in long-lived mammals. On the other hand, a general mechanism for targeting and eliminating cells that behave oddly, compared to most of their neighbors, would presumably be beneficial throughout life (including development), and one could easily imagine it evolving and being maintained by ongoing selection. Consequently, one explanation for the existence of tumor surveillance is that it is the consequence of an existing mechanism that, serendipitously, happens also to enable the elimination of nascent tumor cells. We believe our study supports that novel view.

Among other things, such a view could also explain why tumor surveillance sometimes does and sometimes doesn't occur, depending rather capriciously on the tumor cell genotypes. One might expect a mechanism that evolved for the purpose of tumor surveillance to act on all tumors, whereas that would not necessarily be the case for a mechanism that evolved for other purposes.

Zhu, Wunderlich and Lander

Epithelial cell competition is promoted by signaling from immune cells

RESPONSE to REVIEWER COMMENTS

Response to Reviewer #1

(R1 Major comment 1): 'tumor-suppressive cell competition (TSCC)' instead of 'tumor surveillance'

The authors use of the term 'tumor surveillance' to describe the killing of tumor suppressor gene mutant cells from imaginal discs creates a lot of confusion and misleading to general readers of the article, who may falsely equate the term 'tumor surveillance' with 'tumor immune surveillance'. The term 'tumor surveillance' is already widely used in the mammalian tumor immunology field and involves an active role of circulating immune cells in recognizing and killing tumor cells. For removal of defective cells by non-immune means in mammalian tissue culture, the Fujita lab has used the term Epithelial Defense Against Cancer. In flies, the killing of tumor suppressor gene mutant cells from imaginal discs has usually been called cell competition. Although the Igaki lab once used the term Intrinsic Tumor Suppression, they more recently have carefully chosen 'tumor-suppressive cell competition (TSCC). While it is true that the Vidal lab shows that hemocytes are involved in this, the Bilder lab showed evidence for involvement of cells that are neither immune nor epithelial. The authors themselves are not consistent with its use. At 218, the authors call killing of tumor suppressor gene mutant cells cell competition and lump it together with Myc overexpression competition.

Overall, the authors introduce a new term which further muddies the distinction between the various situations and prevents comparison to the established body of literature. The authors use rhetoric to say in the introduction that fly 'tumor surveillance' and cell competition are thought to be distinct, but then conclude that actually they are not that distinct. Both of these views are overstated 'straw-men'. The authors should say clearly and precisely what the scientific advance is and not equate their findings with 'tumor immune surveillance', which is a different concept entirely. They should revise the manuscript and abstract accordingly.

We appreciate the reviewer's detailed review of different groups' terminology that, as the reviewer notes, has already engendered quite a bit of confusion. It was not our intent to add to that confusion! We had hoped our use of the term "tumor surveillance" would just be taken on face value, i.e., to mean any intrinsic mechanism that detects tumor cells and eliminates them. But we recognize that, for mammalian biologists in particular, this term carries additional baggage of implying the immune system is the primary actor.

Unfortunately, the term "Epithelial Defense Against Cancer" also carries baggage in the other direction—it suggests a mechanism carried out by exclusively by epithelial cells, which is too narrow to describe phenomena in which immune cells play a role.

Although the term "Tumor-suppressive cell competition" has been used by the Igaki lab, widely read papers highlight dissimilarities between classical cell competition and the way

tumor cell clones are eliminated. As one of our goals is to suggest that the two processes are not as dissimilar as some have argued, it would not make sense for us to refer to tumor suppression using a phrase that implies it and cell competition are already known to be one and the same thing.

On balance, we believe the term “Intrinsic Tumor Suppression” is sufficiently neutral in referring to an intrinsic pathway that suppresses tumor cells, without implying what cells are responsible or the mechanism used. In the revised manuscript and abstract, we now use that term throughout, except when we wish to refer to cases in which specific mechanistic assumptions are being invoked (for example we now only use the word “surveillance” when referring to the mammalian case). In addition, we have removed the citation to Igaki et al., on line 218 so that only a cell competition study is cited.

***(R1 Major comment 2)** The Myc model used is atypical and its interpretation is complex. [the authors] do not use a well-defined system for cell competition. This approach by the Gallant lab, used perhaps only once, creates three genotypes of cells, rather than two genotypes which all standard cell competition systems generate. It is for this reason that the Johnston lab in 2014 developed the system that is since almost universally used for Myc competition... In the Gallant lab system three genotypes of cells are created and also three types of borders are created that will differ with each clone. But the competition quantitation measures only the ratio of 2X extra Myc to no extra Myc...is the increase in ratio from excess growth of the 2X myc cells relative to the 1X myc cells, or from apoptosis of the 0x Myc? While this does not make the findings of the paper invalid, it is a limitation in linking the findings to the rest of the literature that uses the Johnston Myc competition system... I request that the authors use the Johnston lab system to test the two main advances claimed: - that hemocytes play a role and that Eiger from hemocytes is important.*

As requested, we performed the additional experiments using the Johnston lab’s approach of generating wildtype clones in a Myc-overexpressing background. The results may be found in Figure S17 and fully support the other results in the manuscript. Specifically, using the wildtype-clones-in-Myc-background approach we confirmed (a) the requirement for hemocytes in cell competition; (b) the requirement for hemocyte *Egr* in cell competition; and (c) the recruitment of hemocytes during cell competition. These experiments also provide new information, such as evidence for dose dependence of hemocyte recruitment on the number of clones per disc, and evidence that disc-derived *Egr* does play a small but significant role in cell competition.

As the major results of this study have now been demonstrated using Myc twin-spot clones, Myc clones in a wildtype background and wildtype clones in a Myc background, as well as Minute twin-spot clones, we hope any remaining concerns have now been allayed.

Figure S1B is a bit misleading because the Minute homozygous cells do not grow and are killed immediately.

We are unclear on what is misleading, as this is the standard approach for demonstrating Minute cell competition. The fact that Minute homozygous cells immediately die in many ways just makes the interpretation easier, as there are only two types of cells to consider.

In Figure 7G: there are many clones that show no CC3 staining, and also the CC3 staining does not show any pattern with respect to clone genotype or the three different types of clonal boundaries. The authors response to Reviewer 2 Major Concern 11 is [thus] insufficient and needs to be quantitated for the system that they use.

As discussed below in response to Major Comment 4, we were also unhappy with the quality of our CC3 staining and have now re-done all of the experiments using an anti-DCP1 antibody instead. The results are much cleaner and we now do see apoptotic cells at clonal boundaries. In Fig. 7F we clearly see them both at 2X-0X boundaries and 2X-1X boundaries and, rarely, at 1X-0X boundaries.

(R1 Major comment 3): No evidence of the functional importance of disc-attached hemocytes

The authors raised concerns about a previous comment (R1 major concern 4), which questioned the functional importance of hemocyte attachment. We appreciate that the authors conducted additional eye-disc washing experiment (Fig S10, S11), which confirm their wing disc observations. However, the authors have not shown the functional importance of disc-attached hemocytes, which undermines the importance of Fig. 2, 3 and 4. Below are our specific comments.

In 'No Wash' condition, the number of disc-attached hemocytes seem to be unchanged between control and Myc conditions. The authors argue that this is due to non-specifically bound hemocytes. Assuming this is the case, the authors cannot argue that hemocytes are 'recruited'. The authors should address this through text changes by toning down the role of 'recruitment' and replace with 'attachment' or 'increased adherence'. Instead, one may interpret these hemocytes as 'stickier' (less detached from disc upon washing).

We have provided additional analysis regarding the disposition of hemocytes associated with discs, including new data in Fig. 2 and Fig. S10. Most importantly, we now show that well-attached hemocytes may be identified, even prior to washing, by their increased nuclear area, a sign of cell spreading. While many hemocytes may loosely adhere to discs when they are first dissected—no surprise given that discs reside in hemolymph—it is clear that there is a subpopulation of tightly attached, flatter hemocytes that appears during cell competition.

Hemocyte attachment and spreading is well known to be associated with hemocyte participation in wound healing and inflammation, a point we now cite. Indeed, others have shown that the mechanism of hemocyte recruitment to wounds in *Drosophila* larvae is increased adhesion and spreading, and not any sort of directed migration (Babcock et al., PNAS 2008), which agrees with what we observe here. As Babcock et al., specifically use the word “recruitment” to refer to such behavior we believe it is appropriate to do so in the present study as well. We now also mention in the manuscript the possibility that

the hemocyte response to cell competition may resemble, in some regards, the hemocyte response to epithelial wounding.

Given the model where hemocytes elevate Egr levels to promote cell competition, it is unclear whether it is only the 'stickier' hemocytes that are the important cells here.

We do not have direct evidence that it is *only* the “stickier” hemocytes that produce Eiger, but we do know that the flattened, well-adhered state is directly associated with hemocyte activation (see above), and hemocyte activation is known to be associated with the production of Eiger (e.g. Fogarty, Caitlin E. et al. 2016, Current Biology, 26(5), 575 – 584). So we certainly think it likely that the specific, well-adhered, flattened hemocytes are the source of Eiger.

2) (Previous comment) Hemocyte attachment on Myc mosaic discs is very variable and some Myc discs have no hemocytes attached.

The authors replied in their rebuttal that '-what we routinely observe after 4-5 washes probably reflects the removal of some specifically bound hemocytes, so we cannot be confident that a disc in which we didn't observe hemocytes lacked any specifically bound ones at the time of dissection'

While it makes sense that additional washes may remove hemocytes, many Myc wing discs have zero or almost zero hemocytes even in 'No Wash' conditions (Fig S10Q: 3/6=50%). Yet, hemocyte ablation consistently (100%) results in a lack of competition (Fig 1G). This observation casts doubt on the interpretation that attachment on discs is indispensable for hemocytes to regulate Myc cell competition.

The authors further replied that '-hemocytes are highly motile cells and likely move on and off discs throughout the experimental period.'

This is an interesting perspective, and it is possible. However, the authors show in detail that a population of hemocytes becomes 'stickier' and attach to the discs even after several washes. It would be expected to see that these 'stickier' hemocytes attach to the disc long-term, so that a change would be quantifiable.

The authors should at least acknowledge the limitations of their interpretation.

The data indicate that *all* hemocytes are removable given enough washing. It would have been much simpler if we could have found a degree of washing that cleanly removes, in every disc, all the non-specifically attached cells and none of the tightly adherent ones, but we were unable to do so. The optimal “signal-to-noise” seems to occur with a degree of washing that removes a significant fraction of the specifically-bound cells. Even then, we cannot be certain every disc gets washed to the same extent as, despite our best efforts, washing is not an exactly reproducible procedure.

Accordingly, the observation that some discs display zero hemocytes (after washing) does not imply that no specifically-bound hemocytes were there initially. The possibility that hemocytes move on and off discs is another potential reason for failing to observe them, but it is not necessary to appeal to that explanation to explain the data.

(R1 Major comment 4): Difficulties in interpreting Caspase-3 as an apoptosis readout
The authors point out that Caspase-3 staining has limitations as a direct readout of

apoptosis. The author's issues are summarized here: 1) hemocyte ablation impacts the clearance of cell debris; 2) the length of clonal boundaries impacts the amount of apoptosis per disc in cell competition context; 3) not all Caspase-3-positive cells undergo cell death; and 4) Myc-overexpression leads to cell-autonomous apoptosis. However, apoptosis measurements are –next to clonal area measurements– one of the essential readouts for cell competition.

Other studies have mastered the listed hardships to produce reliable apoptosis measurements. Each of these issues can be addressed: 1) Apoptotic cells are extruded basally (hemocyte-independent) and the debris is then cleared through phagocytosis by hemocytes. Lolo et al. have faced this issue during their study and observed that the un-cleared debris accumulates specifically on the basal side of the disc, below the epithelial layer. The authors could address this issue by comparing different sections of the disc, e.g. further apical as well as further basal. 2) The authors can address a correlation between clonal boundary length and apoptosis by measuring the length of clonal boundaries and measure adjacent apoptosis, in a similar fashion to the measurements done by Yamamoto et al. previously ('dying cells per clone perimeter', doi.org/10.1038/nature21033). 3) The authors could test alternative readouts of apoptosis, such as TUNEL assay, to avoid non-apoptotic roles of Caspase-3. 4) While cell-autonomous apoptosis is a contributor to the dying cells in discs, this should remain constant across the tested Myc-overexpression contexts in this study.

We appreciate the reviewer's persistence in holding us to a high standard of data collection and presentation. Although we put a lot of effort into distinguishing signal from noise in our activated Caspase 3 staining, we have never been particularly happy with the performance of our antibody and, in response to the reviewer's concerns, we eventually decided to go back to square one, and re-do all of the cell death experiments using a different antibody, which recognizes Dcp1, which is reported to be more effective to detect cell death in *Drosophila* tissues.

We therefore redid all the anti-caspase 3 experiments in original Figures 7, S12, S13, S18, S19, and S20 using anti-Dcp1. Immunostained cells were now much more cleanly visualized, and displayed expected distributions (e.g. adjacent to clonal boundaries)

Consistent with the previous results, the number of Dcp1-positive cells in wing discs undergoing *dMyc*-induced cell competition was increased, and this number went down significantly—similar to that in wing discs carrying control clones—when hemocytes were ablated (See updated Figure 7); either of the two *eiger-RNAi*s was expressed specifically in hemocytes (new Figure S20/original Figure S18), or *bsk* was deleted or *bskRNAi/bsk^{DN}* was expressed specifically in hemocytes (new Figure S21/original Figure S19). Similarly (new Figure S22/original Figure S20), in *Minute* experiments, Dcp1-positive cell numbers decreased significantly following either hemocyte ablation or hemocyte-specific *egr* knock-down.

Given the improvement in sensitivity and specificity provided by the anti-Dcp1 antibody, we believe that the results now stand on their own without need to appeal to explanations based on limitations in the detectability of staining.

(R1 Major comment 5): Imbalanced presentation of preferred model
The text also overweighs the authors' preferred model and does not acknowledge problems in interpretation. As long as the data do not exclude the authors' model, as long as there are possibilities that could explain the differences of this model with published papers, their model is a winner. One example of this is line 279. The results with Minute cells are not statistically significant. The authors provide two explanations why this might be artifactual, but do not state simply that Minute cells may work by a different mechanism.

We have tried to be more balanced in the discussion of models. With regard to the concern about the lack of statistical significance on line 279, since switching to using anti-Dcp-1 to quantify cell death, the results are now statistically significant.

In Figure 3 and response to Reviewer 2. GMR>P35 should not effectively block cell competition in this system –most competition should take place in the proliferating cells ahead of the furrow and not in the SMW behind it. So this genotype does not test whether hemocyte recruitment depends on apoptosis –there will still be extensive competitive apoptosis in the discs. Additionally, any cell blocked by p35 will be 'undead' and recruit hemocytes by a mechanism shown by the Bergmann lab, confusing interpretation.

Regarding the first point, we agree that use of GMR>p35 created difficulties of interpretation and decided to replace these experiments with new ones in which the uniformly expressed arm-Gal4 was used to drive UAS-p35 in the wing disc. The results of these new experiments support the previous conclusions.

Regarding the second point, we do not dispute the fact that expression of p35 in cells in which hid is overexpressed (and thus become “undead”) is accompanied by hemocyte recruitment, as reported by Bergmann. However, one need not interpret such results as indicating that p35 itself, or the “undead” state, recruits hemocytes. Rather, in our view, the simplest interpretation is that hid recruits hemocytes in a manner that is not blocked by p35. The possibility that the activation of pro-apoptotic pathways (upstream of the activated caspase step that is blocked by p35) may well play a central role in hemocyte recruitment seems reasonable; here we seek only to point out that actual cell death is not itself required, a point we now make clearer in the Discussion.

Interestingly, the fact we can also block cell death using a hid mutation, and still not eliminate hemocyte recruitment, suggests that those signals that recruit hemocytes also lie upstream of hid and not just downstream. Since signals such as ROS potentially lie both upstream and downstream of hid (as noted by Bergmann in their 2016 paper), this is perhaps not surprising.

Minor

comments:

(R1 Minor comment 1) 41: the role of systemic factors including Egr and insulin should be included

We have added additional material to the Discussion section to cover this topic.

(R1 Minor comment 2) 116: the authors explain discrepancy with the Moreno lab partly by claiming that their hemocyte elimination technique is more efficient. They show that it

eliminates hemocytes associated with discs, but does it eliminate them from the entire larva? The model proposed apparently does not require hemocytes to be near to the clone.

We do not have a method to quantify the total number of hemocytes in the larva, but what we can easily do is examine discs immediately after removal *without washing*, where we know that a large number of loosely (i.e. nonspecifically) attached hemocytes is typically found. When we do this in *hml>hid* larvae, we detect no *srpHemo-H2A-3xmCherry*-positive cells at all (Figure S3), implying that Hml-positive hemocytes are fairly efficiently eliminated globally. We cannot rule out the possibility that a few remain and, likewise, we cannot say anything about any Hml-negative hemocytes that might be present.

(R1 Minor comment 3) 347-361: *how does this discussion relate to the authors' data that recruitment of hemocytes is not dependent on JNK signaling?*

We apologize for the confusion created by the way this was discussed; this section has been re-written and re-focused in the revised manuscript.

(R1 Minor comment 4) *The pJNK AB in Fig S15 is not convincing. disc A seems to be imaged basally and there is more JNK activation than expected. In B the staining does not seem to correlate with clone genotypes or clone boundaries and there is clear staining in winner as well as loser clones.*

A key transcriptional target of JNK signaling is Puckered (Puc), a Jun kinase phosphatase, which acts in a negative feedback loop to dampen JNK signaling, and we now have introduced *puc-lacZ*, a reporter for JNK signaling, into our *dMyc*-induced cell competition system and stained wing discs of these genotypes with anti- β Gal antibodies instead. Elevated *puc-lacZ* could be observed along clonal boundaries when *dMyc* cell competition was induced, however, this elevation was suppressed when hemocytes were ablated (See new Figure S16, original Figure S15).

(R1 Minor comment 5) 299, *what do the authors mean by the distinction between cell killing and cell competition?*

The observation of cell competition refers to one clone outcompeting another; cell killing is a mechanism that can be responsible (at least in part) for cell competition. In the text we wrote we weren't intending to imply a distinction between cell killing and cell competition, but rather to say that Eiger is required for cell competition, and it is also required specifically for the cell killing that underlies cell competition. We've rephrased the text to try to be clearer.

Response to Reviewer #2

(R2 Major comment 1) *In Figure S20, where they have generated wild-type cells in the Minute background, the caspase 3 staining stains the wild-type cells and not the Minute cells. Also in the same figure when they ablate hemocytes the caspase staining is increased in the wild-type area. These results weakens the conclusions for the Minute competition*

We generated *wild-type* cells in the *Minute* background again, and used anti-Dcp1 antibody to stain dead cells instead of anti-Caspase-3 antibodies. Now the Dcp1 staining

stains the *Minute* cells not the *wild-type* cells. And the Dcp1 staining is decreased instead of being increased when hemocytes are ablated.

Response to Reviewer #3

(R3 Major comment 1) The conclusive remark “These findings support a view of tumor surveillance as less a specific mechanism for eliminating cancer than a general process for promoting phenotypic uniformity among epithelial cells.” is now supported by the data. However, the authors have not fully explored how to interpret the indication from the data in Figure 1 “both dMyc-associated and Minute-associated cell competition requires the presence of hemocytes”, based on the experimental evidence. Mechanistically, although hemocyte-derived Eiger/TNF promotes epithelial cell competition, there is more likely another key signal from hemocytes to explain the strong effect on cell killing in Figure 1.

We would not be surprised to find that something in hemocytes in addition to Eiger is required for cell competition, but don't believe the data provide strong support for that view. It is true that, in our figures, RNAi-knockdown of Eiger has a slightly weaker effect than complete hemocyte ablation, but it is also true that RNAi-mediated knockdown is often incomplete. Given that we believe cell competition operates near a threshold where even small amounts of Eiger (such as the endogenous, low levels in the disc) can trigger weak cell competition (as implied by Fig. S17P), we expect that failure to completely drive Eiger levels to zero in hemocytes would likely manifest as reduced efficacy in blocking cell competition.

This may be beyond the scope of this manuscript but the absence of a novel mechanistic view might make this work less attractive to broader readership.

Primarily, our study does not aim to identify novel mechanisms in cell competition so much as it tries to show that mechanisms underlying cell competition and intrinsic tumor suppression are more similar than previously thought, and even (as we now point out in the discussion) also similar to processes such as wound healing.

These findings suggests that what are widely thought of as cancer-surveillance mechanisms may be better understood as mechanisms evolved for epithelial maintenance and homeostasis—a view we believe will be of considerable interest to a broad readership.

RESPONSE TO REVIEWER COMMENTS (Zhu et al.)

Response to Reviewer #1

The terminology using ‘intrinsic tumor suppression’ is wrong, and this still leads to a straw man model that overstates the distinction between two types of what most people describe as cell competition.

This comment by the reviewer was followed by examples of problematic sections in the text (in the interest of brevity we won’t re-iterate them, but passages specifically pointed out were original lines 59-62 and 80-86 [Introduction], 224-232 [Results], and 327 and 336-337 [Discussion]). Specific changes made to these sections are listed at the end of this response.

We sincerely apologize for not adequately addressing the reviewer’s concerns. The term “intrinsic tumor suppression” was actually introduced to us by the reviewer in the previous round of review, in which our earlier use of the term “tumor surveillance” had been criticized. Apparently, the new term has more historical baggage than we anticipated, and we are happy to abandon it.

We believe the underlying source of conflict here has to do with the fact that we are attempting to address two different audiences: *Drosophila* biologists and cancer researchers. We think the work is important for both fields, but it is especially among cancer researchers—the much larger group—that we feel it has the potential to really change current thinking. Currently, most cancer researchers are committed to the idea that immune-mediated attack of cancer is a cancer-specific process, whereby immune cells specifically recognize cancer cells, typically by the antigens they express. Among the many dozens of reviews on this subject, the idea that immune suppression is a general process for eliminating cells with aberrant phenotypes is almost never raised, even though the implications are profound: If cancer cells are eliminated simply because they *behave* differently from neighbors, and not because they are recognized as *cancer*, it greatly changes thinking about therapeutic intervention.

It might seem curious, given the extensive literature on cell competition in *Drosophila* showing that phenotypic differences alone can drive cell elimination, that cancer researchers haven’t considered this viewpoint. The answer, we believe, has to do with the fact that studies of “classical” (e.g. Myc and Minute) cell competition in *Drosophila* have concluded that immune cells don’t play a central role, while studies of “tumor suppressive” cell competition have concluded they do. So even if a lot of *Drosophila* biologists see this difference as a minor one, and prefer to bundle both processes under a single term, to cancer researchers the difference is a major one, as it supports that bias that elimination of cells by the immune system only occurs with cancer.

We believe this study shatters support for that bias, showing that cell competition involves the immune system in the same way whether it is tumor suppressive or not. To make this clear to cancer researchers we need to highlight the distinction in the *Drosophila* literature concerning when immune cells are, supposedly, directly involved. That distinction clearly exists, so pointing it out is not a “straw man” argument. However, we understand that our terminology might have made it seem we were claiming the existence of an even greater degree of difference. We neither intended, nor need, to do that, and we are happy to revise the text to avoid any unintended impression.

In the new text we have taken care that “cell competition” is used as an overarching term for both tumor-suppressive and non-tumor-suppressive events. Unfortunately, to refer to cell competition that isn’t tumor suppressive, we’ve occasionally had to use constructions we feel are somewhat clumsy, like “classical cell competition” (the word “classical” should be reserved, in our opinion, for science a lot older than this), and had to use terms like “other forms”, “other kinds” or “other examples” of cell competition. Still, we are happy to do it, and we sincerely hope this will resolve any concerns raised. In particular, with regard to the passages specifically called out by the reviewer:

Previous lines 59-62: We now use the phrasing, “In *Drosophila*, similarities between tumor suppression and other examples of cell competition...”

Previous lines 80-86: We re-wrote both this and the preceding paragraph, to avoid making what the reviewer felt was an “exaggerated” distinction. The paragraph is now also shorter and, we think, clearer.

Previous lines 224-232: We thank the author for catching our error in using the same reference (Igaki, 2009) as a citation for the role in Eiger in both classical and tumor-suppressive cell competition. The correct citation for the former should have been Kodra et al., 2024 (reference 58), and the error has now been fixed.

Previous line 327: We have deleted the word “intrinsic”

Previous lines 336-337: We rephrased the sentence so as not to imply an assertion that *scrib* cells are faster growing, but rather simply that they lack the obvious fitness disadvantage characteristic of the slow-growing, loser cells in classical cell competition. We believe this is correct because *scrib* cells do have the capacity for uncontrolled growth that wildtype cells do not.

We have also made changes in several other parts of the manuscript to address the same points and revised the Summary.

(R1 Minor comment 1): *I and the field will also appreciate the authors repeating some experiments with the frequently used WT FLPout in Myc+ background model (S17). They find the elimination of hemocytes to play a significant role here, but it must be said that the effect size is pretty small and much smaller than the effects seen in Fig. 1. I would suggest that the authors make the ‘median’ bars visible in figures like this, they are almost impossible to see, so effect sizes are masked.*

We have added black arrows to point out the ‘median’ bars in Figure S17.

(R1 Minor comment 2): *Fig. S1: I suggest adopting a terminology such as Myc+ to distinguish the extra copies of Myc from the loss of a copy of Minute. The existing figure will suggest to most that a Myc/GFP cell is heterozygous loss of function. Also, thank you for adding in the figure that Minute homozygous cells are killed. As requested, we replaced Myc with Myc+ in Figure S1.*

(R1 Minor comment 3): *is the below accurate? It looks to me that the difference between WT and Hml>egrRNAi in L is smaller than the difference between WT and clone>egrRNAi in P.: “Here we obtained evidence that, in cell competition, Eiger derived from hemocytes plays a major role, but Eiger from loser cells can also contribute to a minor degree (Fig. S17).”*

To make the results clearer, we collected and added more data points into Figure S17. In Figure S17L, the size of *wild-type* clones in a *dMyc* background was markedly and statistically significantly restored toward control values when *egrRNAi* was expressed only in hemocytes.

Response to Reviewer #2

(R2 Major comment 1): *In Figure S22A while the clones (FRT40/FRT40) are obvious, the difference of twin spot (armlacZ, FRT40/ armlacZ, FRT40) versus the unrecombined background (alz FRT40/FRT40) is not obvious.*

We have adjusted the contrast (see updated Figure S22), which we believe makes the difference more distinguishable.

(R2 Major comment 2): *The effect of hml>hid (Figure S22C) and of hml>egrRNAi (Figure S22D) on Dcp1 positive cells is really impressive (Figure S22F), while the effect on area ratio is not so. Why is that?*

Minute mutant cells grow substantially slower than wild type (Martin et al., Development. 2009 doi: 10.1242/dev.038406. PMID: 19855017; see also Figure S4). Even if cell death is entirely blocked, one expects Minute clones to be “outcompeted” to some degree by wild type, explaining the inability to fully rescue area.

(R2 Major comment 3 part 1): *In the same experiment do the authors know if the UAS-hid and hmlGal4 transgenes are on the left or the right arm of the 2nd chromosome. If they are on the left arm, please provide the control experiments (hsFLP; UAS-hid FRT40/ arm-lacZ FRT40) and (hsFLP; hmlGal4 FRT40/ arm-lacZ FRT40) to make sure that the wild type cells (UAS-hid FRT40/UAS-hid FRT40) and (hmlGal4 FRT40/ hmlGal4FRT40) generated in Figure 22C and Figure 22D do not present any growth disadvantage that could interfere with the Minute competition.*

We don't know if the *UAS-hid* and *hml-Gal4* transgenes are on the left or the right arm of the 2nd chromosome, but we have performed the control experiments as requested, inducing control clones on wing discs of *hsFLP; UAS-hid, FRT40A/arm-lacZ, FRT40A* and *hsFLP; hml-Gal4, FRT40A/arm-lacZ, FRT40A* flies. Such clones do not appear to present any growth disadvantage that could interfere with the *Minute* competition (see updated Figure S22).

(R2 Major comment 3 part 2): *In the material and methods the authors state that “Minute ‘twin-spot’ clones were generated as described by Li and Baker65, but using FRT40A instead of FRT42. Larvae carrying transgenes were subjected to a 1-h heat shock at 37°C to create twin-spot clones, and dissected at 96 h ACl.” Why the authors did not perform dissection at 60-72 hrs after heat shock as it is usually performed for the Minute competition. When did they perform the heat shock (how many hours after egg laying)?*

We performed the heat shock 51 h after egg laying.

The authors have provided an experiment showing that hemocyte ablation by hmlGal4>hid does not delay normal development (Figure S4). Have they noticed any delay upon hemocyte depletion when the flies are Minute?

We have now also performed egg-laying assays using *Minute* and *Minute,hml>hid* larvae. We dissected larvae at different time points—108, 120, 132, 144, or 156 h after egg laying, and quantified the posterior compartment size of wing discs stained with anti-Ptc and anti-Wg antibodies. No obvious difference was observed between these two genotypes among all these time points, so there does not seem to be any delay associated with hemocyte depletion in flies that are *Minute* (see updated Figure S4).

Are the dissections in Figure S22A-D performed on the same time after egg laying and heat shock?

The dissections were performed at the same time after heat shock, but they were performed 148 h after egg laying in Figure S22B-D instead of 120 h as in Figure S22A.

Can the authors provide an experiment with hml>hid depletion using the standard conditions that are used for the Minute competition, eg dissection after 60-72 hrs?

As requested, we repeated all the experiments in Figure S22, dissecting larvae 72 h after clone induction. The results are consistent with the previous ones (see updated Figure S22).

(R2 Major comment 4): *In Figure S5A while the clones (FRT40/FRT40) are obvious, the difference of twin spot (armlacZ, FRT40/ armlacZ, FRT40) versus the unrecombined background (alz FRT40/FRT40) is not obvious.*

We have adjusted the contrast (see updated Figure S5), which we believe makes the difference stand out more clearly.

(R2 Minor comment 1):

Since they are different myc competition assays, it would be helpful if in each image was more evident which system was used. For example for Figure S17

B. UAS-myc clones/WT background

C-G. WT clones/ tyb>myc background

I. WT in tub>myc background

J. WT in tub>myc background, hml>hid

K. WT in tub>myc background, hml>egrRNAi

M. WT in tub>myc background

N. egrRNAi in tub>myc background,

O. egrRNAi in tub>myc background, hml>egrRNAi

As requested, we have updated the figure legend of Figure S17.

Response to Reviewer #3

The reviewer commented that the prior revisions addressed the concerns raised by the reviewers to a satisfactory degree. However, the reviewer felt that the title of the manuscript did not reflect the description of the key findings as stated in the summary, and this dampened enthusiasm for the work. We have extensively re-written the summary and hope that the connection is now clearer and satisfactory to the reviewer. As we pointed out in our comments to Reviewer #1, we felt it was important to explicitly “connect the dots” for two different audiences for this work.

RESPONSE TO REVIEWER COMMENTS

Response to Reviewer #1

Reviewer #1 (Remarks to the Author):

Thanks to the authors for engaging thoughtfully with the critique, which i think has clarified the paper's contribution.

- *One detail I noticed in Fig. 3c: I was not aware that heterozygosity for hid alone, rather than the H99 deficiency that removes multiple apoptotic regulators, was sufficient to suppress cell death. The authors may want to add a reference that supports that, and also make clear at the top panel that this is hid [1]/+.*

As requested, we have updated the top panel of Figure 3 and added the corresponding reference to the main text. According to the reference hid¹ is actually an antimorphic allele, which likely explains why it has a relatively strong effect when heterozygous.

Response to Reviewer #2

Reviewer #2 (Remarks to the Author):

I would like to thank the authors for their responses.

- *Major concern:*

*1) In control experiments of panel S22A, S22B, S22C, S22G, 22H and 22I the twin spots are not obvious, while they should be present and have similar size with clones.
2) In the experiment provided in 22K and 22L images, the staining for the clones (b-galactosidase) presents 3 different cell types (here we expect only 2). One does not have lacZ (wild type cells), one has lower lacZ staining (Rp/+ cells) and the third cell type seems brighter (like having two copies of lacZ). We do not expect cells with 2 copies of lacZ to be viable, since they lack both Rp copies.
Authors should provide a better disc for the these panels.*

As requested, we have updated panels A, B, C, G, H, I, K, and L of Figure S22 to provide images that better match the expectations for brightness levels. It should be noted, however, that images we presented in Figure S22 are maximum projections of Z-stacks, therefore, absolute brightness may vary in ways that are not entirely related to number of copies of LacZ.

I would prefer the authors to tone down their conclusions for Minute competition or at least mention that only one Minute mutation was studied.

As requested, we now specifically mention that only one *Minute* mutation was used in this study.